# Multi-Agent Reinforcement Learning via Double Averaging Primal-Dual Optimization

**Hoi-To Wai**
The Chinese University of Hong Kong
Shatin, Hong Kong
htwai@se.cuhk.edu.hk

**Zhuoran Yang**
Princeton University
Princeton, NJ, USA
zy6@princeton.edu

**Zhaoran Wang**
Northwestern University
Evanston, IL, USA
zhaoranwang@gmail.com

**Mingyi Hong**
University of Minnesota
Minneapolis, MN, USA
mhong@umn.edu

## Abstract

Despite the success of single-agent reinforcement learning, multi-agent reinforcement learning (MARL) remains challenging due to complex interactions between agents. Motivated by decentralized applications such as sensor networks, swarm robotics, and power grids, we study policy evaluation in MARL, where agents with jointly observed state-action pairs and private local rewards collaborate to learn the value of a given policy.

In this paper, we propose a double averaging scheme, where each agent iteratively performs averaging over both space and time to incorporate neighboring gradient information and local reward information, respectively. We prove that the proposed algorithm converges to the optimal solution at a global geometric rate. In particular, such an algorithm is built upon a primal-dual reformulation of the mean squared projected Bellman error minimization problem, which gives rise to a decentralized convex-concave saddle-point problem. To the best of our knowledge, the proposed double averaging primal-dual optimization algorithm is the first to achieve fast finite-time convergence on decentralized convex-concave saddle-point problems.

## 1 Introduction

Reinforcement learning combined with deep neural networks recently achieves superhuman performance on various challenging tasks such as video games and board games [34, 45]. In these tasks, an agent uses deep neural networks to learn from the environment and adaptively makes optimal decisions. Despite the success of single-agent reinforcement learning, multi-agent reinforcement learning (MARL) remains challenging, since each agent interacts with not only the environment but also other agents. In this paper, we study collaborative MARL with local rewards. In this setting, all the agents share a joint state whose transition dynamics is determined together by the local actions of individual agents. However, each agent only observes its own reward, which may differ from that of other agents. The agents aim to collectively maximize the global sum of local rewards. To collaboratively make globally optimal decisions, the agents need to exchange local information. Such a setting of MARL is ubiquitous in large-scale applications such as sensor networks [42, 9], swarm robotics [23, 8], and power grids [3, 13].

A straightforward idea is to set up a central node that collects and broadcasts the reward information, and assigns the action of each agent. This reduces the multi-agent problem into a single-agent one. However, the central node is often unscalable, susceptible to malicious attacks, and even infeasible

in large-scale applications. Moreover, such a central node is a single point of failure, which is susceptible to adversarial attacks. In addition, the agents are likely to be reluctant to reveal their local reward information due to privacy concerns [5, 27], which makes the central node unattainable.

To make MARL more scalable and robust, we propose a decentralized scheme for exchanging local information, where each agent only communicates with its neighbors over a network. In particular, we study the policy evaluation problem, which aims to learn a global value function of a given policy. We focus on minimizing a Fenchel duality-based reformulation of the mean squared Bellman error in the model-free setting with infinite horizon, batch trajectory, and linear function approximation.

At the core of the proposed algorithm is a "double averaging" update scheme, in which the algorithm performs one average over space (across agents to ensure consensus) and one over time (across observations along the trajectory). In detail, each agent locally tracks an estimate of the full gradient and incrementally updates it using two sources of information: (i) the stochastic gradient evaluated on a new pair of joint state and action along the trajectory and the corresponding local reward, and (ii) the local estimates of the full gradient tracked by its neighbors. Based on the updated estimate of the full gradient, each agent then updates its local copy of the primal parameter. By iteratively propagating the local information through the network, the agents reach global consensus and collectively attain the desired primal parameter, which gives an optimal approximation of the global value function.

**Related Work** The study of MARL in the context of Markov game dates back to [28]. See also [29, 24, 21] and recent works on collaborative MARL [51, 1]. However, most of these works consider the tabular setting, which suffers from the curse of dimensionality. To address this issue, under the collaborative MARL framework, [53] and [25] study actor-critic algorithms and policy evaluation with on linear function approximation, respectively. However, their analysis is asymptotic in nature and largely relies on two-time-scale stochastic approximation using ordinary differential equations [2], which is tailored towards the continuous-time setting. Meanwhile, most works on collaborative MARL impose the simplifying assumption that the local rewards are identical across agents, making it unnecessary to exchange the local information. More recently, [17–19, 31, 37] study deep MARL that uses deep neural networks as function approximators. However, most of these works focus on empirical performance and lack theoretical guarantees. Also, they do not emphasize on the efficient exchange of information across agents. In addition to MARL, another line of related works study multi-task reinforcement learning (MTRL), in which an agent aims to solve multiple reinforcement learning problems with shared structures [52, 39, 32, 33, 48].

The primal-dual formulation of reinforcement learning is studied in [30, 32, 33, 26, 10, 7, 50, 12, 11, 15] among others. Except for [32, 33] discussed above, most of these works study the single-agent setting. Among them, [26, 15] are most related to our work. In specific, they develop variance reduction-based algorithms [22, 14, 43] to achieve the geometric rate of convergence in the setting with batch trajectory. In comparison, our algorithm is based on the aforementioned double averaging update scheme, which updates the local estimates of the full gradient using both the estimates of neighbors and new states, actions, and rewards. In the single-agent setting, our algorithm is closely related to stochastic average gradient (SAG) [43] and stochastic incremental gradient (SAGA) [14], with the difference that our objective function is a finite sum convex-concave saddle-point problem. Our work is also related to prior work in the broader contexts of primal-dual and multi-agent optimization. For example, [38] apply variance reduction techniques to convex-concave saddle-point problems to achieve the geometric rate of convergence. However, their algorithm is centralized and it is unclear whether their approach is readily applicable to the multi-agent setting. Another line of related works study multi-agent optimization, for example, [49, 36, 6, 44, 41]. However, these works mainly focus on the general setting where the objective function is a sum of convex local cost functions. To the best of our knowledge, our work is the first to address decentralized convex-concave saddle-point problems with sampled observations that arise from MARL.

**Contribution** Our contribution is threefold: (i) We reformulate the multi-agent policy evaluation problem using Fenchel duality and propose a decentralized primal-dual optimization algorithm with a double averaging update scheme. (ii) We establish the global geometric rate of convergence for the proposed algorithm, making it the first algorithm to achieve fast linear convergence for MARL. (iii) Our proposed algorithm and analysis is of independent interest for solving a broader class of decentralized convex-concave saddle-point problems with sampled observations.

**Organization** In §2 we introduce the problem formulation of MARL. In §3 we present the proposed algorithm and lay out the convergence analysis. In §4 we illustrate the empirical performance of the proposed algorithm. We defer the detailed proofs to the supplementary material.

**Notation** Unless otherwise specified, for a vector $\boldsymbol{x}$, $\|\boldsymbol{x}\|$ denotes its Euclidean norm; for a matrix $\boldsymbol{X}$, $\|\boldsymbol{X}\|$ denotes its spectral norm, *i.e.,* the largest singular value.

## 2  Problem Formulation

In this section, we introduce the background of MARL, which is modeled as a multi-agent Markov decision process (MDP). Under this model, we formulate the policy evaluation problem as a primal-dual convex-concave optimization problem.

**Multi-agent MDP** Consider a group of $N$ agents. We are interested in the multi-agent MDP:

$$\left(\mathcal{S}, \{\mathcal{A}_i\}_{i=1}^N, \mathcal{P}^{\boldsymbol{a}}, \{\mathcal{R}_i\}_{i=1}^N, \gamma\right),$$

where $\mathcal{S}$ is the state space and $\mathcal{A}_i$ is the action space for agent $i$. We write $\boldsymbol{s} \in \mathcal{S}$ and $\boldsymbol{a} := (a_1, ..., a_N) \in \mathcal{A}_1 \times \cdots \times \mathcal{A}_N$ as the joint state and action, respectively. The function $\mathcal{R}_i(\boldsymbol{s}, \boldsymbol{a})$ is the local reward received by agent $i$ after taking joint action $\boldsymbol{a}$ at state $\boldsymbol{s}$, and $\gamma \in (0, 1)$ is the discount factor. Both $\boldsymbol{s}$ and $\boldsymbol{a}$ are available to all agents, whereas the reward $\mathcal{R}_i$ is *private* for agent $i$.

In contrast to a single-agent MDP, the agents are coupled together by the state transition matrix $\mathcal{P}^{\boldsymbol{a}} \in \mathbb{R}^{|\mathcal{S}| \times |\mathcal{S}|}$, whose $(\boldsymbol{s}, \boldsymbol{s}')$-th element is the probability of transiting from $\boldsymbol{s}$ to $\boldsymbol{s}'$, after taking a joint action $\boldsymbol{a}$. This scenario arises from large-scale applications such as sensor networks [42, 9], swarm robotics [23, 8], and power grids [3, 13], which strongly motivates the development of a multi-agent RL strategy. Moreover, under the collaborative setting, the goal is to maximize the collective return of all agents. Suppose there exists a central controller that collects the rewards of, and assigns the action to each individual agent, the problem reduces to the classical MDP with action space $\mathcal{A}$ and global reward function $R_c(\boldsymbol{s}, \boldsymbol{a}) = N^{-1} \sum_{i=1}^N \mathcal{R}_i(\boldsymbol{s}, \boldsymbol{a})$. Thus, without such a central controller, it is essential for the agents to collaborate with each other so as to solve the multi-agent problem based solely on local information.

Furthermore, a joint policy, denoted by $\boldsymbol{\pi}$, specifies the rule of making sequential decisions for the agents. Specifically, $\boldsymbol{\pi}(\boldsymbol{a}|\boldsymbol{s})$ is the conditional probability of taking joint action $\boldsymbol{a}$ given the current state $\boldsymbol{s}$. We define the reward function of joint policy $\boldsymbol{\pi}$ as an average of the local rewards:

$$R_c^{\boldsymbol{\pi}}(\boldsymbol{s}) := \tfrac{1}{N} \sum_{i=1}^N R_i^{\boldsymbol{\pi}}(\boldsymbol{s}), \qquad \text{where } R_i^{\boldsymbol{\pi}}(\boldsymbol{s}) := \mathbb{E}_{\boldsymbol{a} \sim \boldsymbol{\pi}(\cdot|\boldsymbol{s})}\left[\mathcal{R}_i(\boldsymbol{s}, \boldsymbol{a})\right]. \tag{1}$$

That is, $R_c^{\boldsymbol{\pi}}(\boldsymbol{s})$ is the expected value of the average of the rewards when the agents follow policy $\boldsymbol{\pi}$ at state $\boldsymbol{s}$. Besides, any fixed policy $\boldsymbol{\pi}$ induces a Markov chain over $\mathcal{S}$, whose transition matrix is denoted by $\boldsymbol{P}^{\boldsymbol{\pi}}$. The $(\boldsymbol{s}, \boldsymbol{s}')$-th element of $\boldsymbol{P}^{\boldsymbol{\pi}}$ is given by

$$[\boldsymbol{P}^{\boldsymbol{\pi}}]_{\boldsymbol{s}, \boldsymbol{s}'} = \sum_{\boldsymbol{a} \in \mathcal{A}} \boldsymbol{\pi}(\boldsymbol{a}|\boldsymbol{s}) \cdot [\mathcal{P}^{\boldsymbol{a}}]_{\boldsymbol{s}, \boldsymbol{s}'}.$$

When this Markov chain is aperiodic and irreducible, it induces a stationary distribution $\mu^{\boldsymbol{\pi}}$ over $\mathcal{S}$.

**Policy Evaluation** A central problem in reinforcement learning is *policy evaluation*, which refers to learning the *value function* of a given policy. This problem appears as a key component in both value-based methods such as policy iteration, and policy-based methods such as actor-critic algorithms [46]. Thus, efficient estimation of the value functions in multi-agent MDPs enables us to extend the successful approaches in single-agent RL to the setting of MARL.

Specifically, for any given joint policy $\boldsymbol{\pi}$, the value function of $\boldsymbol{\pi}$, denoted by $V^{\boldsymbol{\pi}} : \mathcal{S} \to \mathbb{R}$, is defined as the expected value of the discounted cumulative reward when the multi-agent MDP is initialized with a given state and the agents follows policy $\boldsymbol{\pi}$ afterwards. For any state $\boldsymbol{s} \in \mathcal{S}$, we define

$$V^{\boldsymbol{\pi}}(\boldsymbol{s}) := \mathbb{E}\left[\sum_{p=1}^{\infty} \gamma^p \mathcal{R}_c^{\boldsymbol{\pi}}(\boldsymbol{s}_p) | \boldsymbol{s}_1 = \boldsymbol{s}, \boldsymbol{\pi}\right]. \tag{2}$$

To simplify the notation, we define the vector $\boldsymbol{V}^{\boldsymbol{\pi}} \in \mathbb{R}^{|\mathcal{S}|}$ through stacking up $V^{\boldsymbol{\pi}}(\boldsymbol{s})$ in (2) for all $\boldsymbol{s}$. By definition, $\boldsymbol{V}^{\boldsymbol{\pi}}$ satisfies the Bellman equation

$$\boldsymbol{V}^{\boldsymbol{\pi}} = \boldsymbol{R}_c^{\boldsymbol{\pi}} + \gamma \boldsymbol{P}^{\boldsymbol{\pi}} \boldsymbol{V}^{\boldsymbol{\pi}}, \tag{3}$$

where $\boldsymbol{R}_c^{\boldsymbol{\pi}}$ is obtained by stacking up (1) and $[\boldsymbol{P}^{\boldsymbol{\pi}}]_{\boldsymbol{s}, \boldsymbol{s}'} := \mathbb{E}_{\boldsymbol{\pi}}[\mathcal{P}^{\boldsymbol{a}}_{\boldsymbol{s}, \boldsymbol{s}'}]$ is the expected transition matrix. Moreover, it can be shown that $\boldsymbol{V}^{\boldsymbol{\pi}}$ is the unique solution of (3).

When the number of states is large, it is impossible to store $\boldsymbol{V}^{\boldsymbol{\pi}}$. Instead, our goal is to learn an approximate version of the value function via function approximation. In specific, we approximate $V^{\boldsymbol{\pi}}(\boldsymbol{s})$ using the family of linear functions

$$\left\{V_{\boldsymbol{\theta}}(\boldsymbol{s}) := \boldsymbol{\phi}^{\top}(\boldsymbol{s})\boldsymbol{\theta} : \boldsymbol{\theta} \in \mathbb{R}^d\right\},$$

where $\boldsymbol{\theta} \in \mathbb{R}^d$ is the parameter, $\boldsymbol{\phi}(\boldsymbol{s}) \colon \mathcal{S} \to \mathbb{R}^d$ is a known dictionary consisting of $d$ features, e.g., a feature mapping induced by a neural network. To simplify the notation, we define $\boldsymbol{\Phi} := (\dots; \boldsymbol{\phi}^\top(\boldsymbol{s}); \dots) \in \mathbb{R}^{|\mathcal{S}| \times d}$ and let $\boldsymbol{V_\theta} \in \mathbb{R}^{|\mathcal{S}|}$ be the vector constructed by stacking up $\{V_{\boldsymbol{\theta}}(\boldsymbol{s})\}_{\boldsymbol{s} \in \mathcal{S}}$.

With function approximation, our problem is reduced to finding a $\boldsymbol{\theta} \in \mathbb{R}^d$ such that $\boldsymbol{V_\theta} \approx \boldsymbol{V^\pi}$. Specifically, we seek for $\boldsymbol{\theta}$ such that the mean squared projected Bellman error (MSPBE)

$$\mathsf{MSPBE}^\star(\boldsymbol{\theta}) := \frac{1}{2} \left\| \boldsymbol{\Pi_\Phi} \left( \boldsymbol{V_\theta} - \gamma \boldsymbol{P^\pi} \boldsymbol{V_\theta} - \boldsymbol{R_c^\pi} \right) \right\|_{\boldsymbol{D}}^2 + \rho \|\boldsymbol{\theta}\|^2 \tag{4}$$

is minimized, where $\boldsymbol{D} = \mathrm{diag}[\{\mu^{\boldsymbol{\pi}}(\boldsymbol{s})\}_{\boldsymbol{s} \in \mathcal{S}}] \in \mathbb{R}^{|\mathcal{S}| \times |\mathcal{S}|}$ is a diagonal matrix constructed using the stationary distribution of $\boldsymbol{\pi}$, $\boldsymbol{\Pi_\Phi} \colon \mathbb{R}^{|\mathcal{S}|} \to \mathbb{R}^{|\mathcal{S}|}$ is the projection onto subspace $\{\boldsymbol{\Phi\theta} \colon \boldsymbol{\theta} \in \mathbb{R}^d\}$, defined as $\boldsymbol{\Pi_\Phi} = \boldsymbol{\Phi}(\boldsymbol{\Phi^\top D \Phi})^{-1}\boldsymbol{\Phi^\top D}$, and $\rho \geq 0$ is a free parameter controlling the regularization on $\boldsymbol{\theta}$. For any positive semidefinite matrix $\boldsymbol{A}$, we define $\|\boldsymbol{v}\|_{\boldsymbol{A}} = \sqrt{\boldsymbol{v}^\top \boldsymbol{A} \boldsymbol{v}}$ for any vector $\boldsymbol{v}$. By direct computation, when $\boldsymbol{\Phi^\top D \Phi}$ is invertible, the MSPBE defined in (4) can be written as

$$\mathsf{MSPBE}^\star(\boldsymbol{\theta}) = \frac{1}{2} \left\| \boldsymbol{\Phi^\top D} \left( \boldsymbol{V_\theta} - \gamma \boldsymbol{P^\pi} \boldsymbol{V_\theta} - \boldsymbol{R_c^\pi} \right) \right\|_{(\boldsymbol{\Phi^\top D \Phi})^{-1}}^2 + \rho \|\boldsymbol{\theta}\|^2 = \frac{1}{2} \left\| \boldsymbol{A\theta} - \boldsymbol{b} \right\|_{\boldsymbol{C}^{-1}}^2 + \rho \|\boldsymbol{\theta}\|^2, \tag{5}$$

where we define $\boldsymbol{A} := \mathbb{E}\big[\boldsymbol{\phi}(\boldsymbol{s}_p)\big(\boldsymbol{\phi}(\boldsymbol{s}_p) - \gamma\boldsymbol{\phi}(\boldsymbol{s}_{p+1})\big)^\top\big]$, $\boldsymbol{C} := \mathbb{E}\big[\boldsymbol{\phi}(\boldsymbol{s}_p)\boldsymbol{\phi}^\top(\boldsymbol{s}_p)\big]$, and $\boldsymbol{b} := \mathbb{E}\big[\mathcal{R}_c^{\boldsymbol{\pi}}(\boldsymbol{s}_p)\boldsymbol{\phi}(\boldsymbol{s}_p)\big]$. Here the expectations in $\boldsymbol{A}$, $\boldsymbol{b}$, and $\boldsymbol{C}$ are all taken with respect to (w.r.t.) the stationary distribution $\mu^{\boldsymbol{\pi}}$. Furthermore, when $\boldsymbol{A}$ is full rank and $\boldsymbol{C}$ is positive definite, it can be shown that the MSPBE in (5) has a unique minimizer.

To obtain a practical optimization problem, we replace the expectations above by their sampled averages from $M$ samples. In specific, for a given policy $\boldsymbol{\pi}$, a finite state-action sequence $\{\boldsymbol{s}_p, \boldsymbol{a}_p\}_{p=1}^M$ is simulated from the multi-agent MDP using joint policy $\boldsymbol{\pi}$. We also observe $\boldsymbol{s}_{M+1}$, the next state of $\boldsymbol{s}_M$. Then we construct the sampled versions of $\boldsymbol{A}, \boldsymbol{b}, \boldsymbol{C}$, denoted respectively by $\hat{\boldsymbol{A}}, \hat{\boldsymbol{C}}, \hat{\boldsymbol{b}}$, as

$$\begin{aligned}
&\hat{\boldsymbol{A}} := \tfrac{1}{M} \textstyle\sum_{p=1}^M \boldsymbol{A}_p, \ \hat{\boldsymbol{C}} := \tfrac{1}{M} \textstyle\sum_{p=1}^M \boldsymbol{C}_p, \ \hat{\boldsymbol{b}} := \tfrac{1}{M} \textstyle\sum_{p=1}^M \boldsymbol{b}_p, \ \text{with} \\
&\boldsymbol{A}_p := \boldsymbol{\phi}(\boldsymbol{s}_p)\big(\boldsymbol{\phi}(\boldsymbol{s}_p) - \gamma\boldsymbol{\phi}(\boldsymbol{s}_{p+1})\big)^\top, \ \boldsymbol{C}_p := \boldsymbol{\phi}(\boldsymbol{s}_p)\boldsymbol{\phi}^\top(\boldsymbol{s}_p), \ \boldsymbol{b}_p := \mathcal{R}_c(\boldsymbol{s}_p, \boldsymbol{a}_p)\boldsymbol{\phi}(\boldsymbol{s}_p),
\end{aligned} \tag{6}$$

where $\mathcal{R}_c(\boldsymbol{s}_p, \boldsymbol{a}_p) := N^{-1} \sum_{i=1}^N \mathcal{R}_i(\boldsymbol{s}_p, \boldsymbol{a}_p)$ is the average of the local rewards received by each agent when taking action $\boldsymbol{a}_p$ at state $\boldsymbol{s}_p$. Here we assume that $M$ is sufficiently large such that $\hat{\boldsymbol{C}}$ is invertible and $\hat{\boldsymbol{A}}$ is full rank. Using the terms defined in (6), we obtain the empirical MSPBE

$$\mathsf{MSPBE}(\boldsymbol{\theta}) := \frac{1}{2} \left\| \hat{\boldsymbol{A}}\boldsymbol{\theta} - \hat{\boldsymbol{b}} \right\|_{\hat{\boldsymbol{C}}^{-1}}^2 + \rho \|\boldsymbol{\theta}\|^2, \tag{7}$$

which converges to $\mathsf{MSPBE}^\star(\boldsymbol{\theta})$ as $M \to \infty$. Let $\hat{\boldsymbol{\theta}}$ be a minimizer of the empirical MSPBE, our estimation of $\boldsymbol{V^\pi}$ is given by $\boldsymbol{\Phi}\hat{\boldsymbol{\theta}}$. Since the rewards $\{\mathcal{R}_i(\boldsymbol{s}_p, \boldsymbol{a}_p)\}_{i=1}^N$ are private to each agent, it is impossible for any agent to compute $\mathcal{R}_c(\boldsymbol{s}_p, \boldsymbol{a}_p)$, and minimize the empirical MSPBE (7) independently.

**Multi-agent, Primal-dual, Finite-sum Optimization** Recall that under the multi-agent MDP, the agents are able to observe the states and the joint actions, but can only observe their local rewards. Thus, each agent is able to compute $\hat{\boldsymbol{A}}$ and $\hat{\boldsymbol{C}}$ defined in (6), but is unable to obtain $\hat{\boldsymbol{b}}$. To resolve this issue, for any $i \in \{1, \dots, N\}$ and any $p \in \{1, \dots, M\}$, we define $\boldsymbol{b}_{p,i} := \mathcal{R}_i(\boldsymbol{s}_p, \boldsymbol{a}_p)\boldsymbol{\phi}(\boldsymbol{s}_p)$ and $\hat{\boldsymbol{b}}_i := M^{-1}\sum_{p=1}^M \boldsymbol{b}_{p,i}$, which are known to agent $i$ only. By direct computation, it is easy to verify that minimizing $\mathsf{MSPBE}(\boldsymbol{\theta})$ in (7) is equivalent to solving

$$\min_{\boldsymbol{\theta} \in \mathbb{R}^d} \frac{1}{N} \sum_{i=1}^N \mathsf{MSPBE}_i(\boldsymbol{\theta}) \ \text{ where } \ \mathsf{MSPBE}_i(\boldsymbol{\theta}) := \frac{1}{2} \left\| \hat{\boldsymbol{A}}\boldsymbol{\theta} - \hat{\boldsymbol{b}}_i \right\|_{\hat{\boldsymbol{C}}^{-1}}^2 + \rho \|\boldsymbol{\theta}\|^2. \tag{8}$$

The equivalence can be seen by comparing the optimality conditions of two optimization problems.

Importantly, (8) is a *multi-agent optimization problems* [36] whose objective is to minimize a summation of $N$ local functions coupled together by the common parameter $\boldsymbol{\theta}$. Here $\mathsf{MSPBE}_i(\boldsymbol{\theta})$ is private to agent $i$ and the parameter $\boldsymbol{\theta}$ is shared by all agents. As inspired by [35, 30, 15], using Fenchel duality, we obtain the conjugate form of $\mathsf{MSPBE}_i(\boldsymbol{\theta})$, i.e.,

$$\frac{1}{2} \left\| \hat{\boldsymbol{A}}\boldsymbol{\theta} - \hat{\boldsymbol{b}}_i \right\|_{\hat{\boldsymbol{C}}^{-1}}^2 + \rho \|\boldsymbol{\theta}\|^2 = \max_{\boldsymbol{w}_i \in \mathbb{R}^d} \left( \boldsymbol{w}_i^\top \big(\hat{\boldsymbol{A}}\boldsymbol{\theta} - \hat{\boldsymbol{b}}_i\big) - \frac{1}{2} \boldsymbol{w}_i^\top \hat{\boldsymbol{C}} \boldsymbol{w}_i \right) + \rho \|\boldsymbol{\theta}\|^2. \tag{9}$$

Observe that each of $\hat{A}, \hat{C}, \hat{b}_i$ can be expressed as a finite sum of matrices/vectors. By (9), problem (8) is equivalent to a *multi-agent*, *primal-dual* and *finite-sum* optimization problem:

$$\min_{\boldsymbol{\theta} \in \mathbb{R}^d} \max_{\boldsymbol{w}_i \in \mathbb{R}^d, i=1,\dots,N} \frac{1}{NM} \sum_{i=1}^{N} \sum_{p=1}^{M} \underbrace{\left( \boldsymbol{w}_i^\top \boldsymbol{A}_p \boldsymbol{\theta} - \boldsymbol{b}_{p,i}^\top \boldsymbol{w}_i - \frac{1}{2} \boldsymbol{w}_i^\top \boldsymbol{C}_p \boldsymbol{w}_i + \frac{\rho}{2} \|\boldsymbol{\theta}\|^2 \right)}_{:= J_{i,p}(\boldsymbol{\theta}, \boldsymbol{w}_i)} . \quad (10)$$

Hereafter, the global objective function is denoted by $J(\boldsymbol{\theta}, \{\boldsymbol{w}_i\}_{i=1}^{N}) := (1/NM) \sum_{i=1}^{N} \sum_{p=1}^{M} J_{i,p}(\boldsymbol{\theta}, \boldsymbol{w}_i)$, which is convex *w.r.t.* the primal variable $\boldsymbol{\theta}$ and is concave *w.r.t.* the dual variable $\{\boldsymbol{w}_i\}_{i=1}^{N}$.

It is worth noting that the challenges in solving (10) are three-fold. First, to obtain a saddle-point solution $(\{\boldsymbol{w}_i\}_{i=1}^{N}, \boldsymbol{\theta})$, any algorithm for (10) needs to update the primal and dual variables simultaneously, which can be difficult as objective function needs not be strongly convex with respect to $\boldsymbol{\theta}$. In this case, it is nontrivial to compute a solution efficiently. Second, the objective function of (10) consists of a sum of $M$ functions, with $M \gg 1$ potentially, such that conventional primal-dual methods [4] can no longer be applied due to the increased complexity. Lastly, since $\boldsymbol{\theta}$ is shared by all the agents, when solving (10), the $N$ agents need to reach a consensus on $\boldsymbol{\theta}$ without sharing the local functions, e.g., $J_{i,p}(\cdot)$ has to remain unknown to all agents except for agent $i$ due to privacy concerns. Although finite-sum convex optimization problems with shared variables are well-studied, new algorithms and theory are needed for convex-concave saddle-point problems. Next, we propose a novel decentralized first-order algorithm that tackles these difficulties and converges to a saddle-point solution of (10) with linear rate.

## 3  Primal-dual Distributed Incremental Aggregated Gradient Method

We are ready to introduce our algorithm for solving the optimization problem in (10). Since $\boldsymbol{\theta}$ is shared by all the $N$ agents, the agents need to exchange information so as to reach a consensual solution. Let us first specify the communication model. We assume that the $N$ agents communicate over a network specified by a connected and undirected graph $G = (V, E)$, with $V = [N] = \{1, ..., N\}$ and $E \subseteq V \times V$ being its vertex set and edge set, respectively. Over $G$, it is possible to define a doubly stochastic matrix $\boldsymbol{W}$ such that $W_{ij} = 0$ if $(i,j) \notin E$ and $\boldsymbol{W}\mathbf{1} = \boldsymbol{W}^\top \mathbf{1} = \mathbf{1}$, note $\lambda := \lambda_{\mathsf{max}}(\boldsymbol{W} - N^{-1}\mathbf{1}\mathbf{1}^\top) < 1$ since $G$ is connected. Notice that the edges in $G$ may be formed independently of the coupling between agents in the MDP induced by the stochastic policy $\boldsymbol{\pi}$.

We handle problem (10) by judiciously combining the techniques of *dynamic consensus* [41, 54] and *stochastic (or incremental) average gradient* (SAG) [20, 43], which have been developed independently in the control and machine learning communities, respectively. From a high level viewpoint, our method utilizes a gradient estimator which tracks the gradient over *space* (across $N$ agents) and *time* (across $M$ samples). To proceed with our development while explaining the intuitions, we first investigate a centralized and batch algorithm for solving (10).

**Centralized Primal-dual Optimization**  Consider the primal-dual gradient updates. For any $t \geq 1$, at the $t$-th iteration, we update the primal and dual variables by

$$\boldsymbol{\theta}^{t+1} = \boldsymbol{\theta}^t - \gamma_1 \nabla_{\boldsymbol{\theta}} J(\boldsymbol{\theta}^t, \{\boldsymbol{w}_i^t\}_{i=1}^{N}), \qquad \boldsymbol{w}_i^{t+1} = \boldsymbol{w}_i^t + \gamma_2 \nabla_{\boldsymbol{w}_i} J(\boldsymbol{\theta}^t, \{\boldsymbol{w}_i^t\}_{i=1}^{N}), \, i \in [N] , \quad (11)$$

where $\gamma_1, \gamma_2 > 0$ are step sizes, which is a simple application of a gradient descent/ascent update to the primal/dual variables. As shown by Du et al. [15], when $\hat{A}$ is full rank and $\hat{C}$ is invertible, the Jacobian matrix of the primal-dual optimal condition is full rank. Thus, within a certain range of step size $(\gamma_1, \gamma_2)$, recursion (11) converges linearly to the optimal solution of (10).

**Proposed Method**  The primal-dual gradient method in (11) serves as a reasonable template for developing an efficient decentralized algorithm for (10). Let us focus on the update of the primal variable $\boldsymbol{\theta}$ in (11), which is a more challenging part since $\boldsymbol{\theta}$ is shared by all $N$ agents. To evaluate the gradient *w.r.t.* $\boldsymbol{\theta}$, we observe that – (a) agent $i$ does not have access to the functions, $\{J_{j,p}(\cdot), j \neq i\}$, of the other agents; (b) computing the gradient requires summing up the contributions from $M$ samples. As $M \gg 1$, doing so is undesirable since the computation complexity would be $\mathcal{O}(Md)$.

We circumvent the above issues by utilizing a *double gradient tracking* scheme for the primal $\boldsymbol{\theta}$-update and an incremental update scheme for the local dual $\boldsymbol{w}_i$-update in the following primal-dual distributed incremental aggregated gradient (PD-DistIAG) method. Here each agent $i \in [N]$

**Algorithm 1** PD-DistIAG **Method** for Multi-agent, Primal-dual, Finite-sum Optimization

---

**Input**: Initial estimators $\{\boldsymbol{\theta}_i^1, \boldsymbol{w}_i^1\}_{i \in [N]}$, initial gradient estimators $\boldsymbol{s}_i^0 = \boldsymbol{d}_i^0 = \boldsymbol{0}, \forall\, i \in [N]$, initial counter $\tau_p^0 = 0, \forall\, p \in [M]$, and stepsizes $\gamma_1, \gamma_2 > 0$.
**for** $t \geq 1$ **do**

   The agents pick a common sample indexed by $p_t \in \{1, ..., M\}$.
   Update the counter variable as:
$$\tau_{p_t}^t = t, \;\; \tau_p^t = \tau_p^{t-1}, \; \forall\, p \neq p_t \;. \tag{12}$$

   **for** each agent $i \in \{1, \ldots, N\}$ **do**
      Update the gradient surrogates by
$$\boldsymbol{s}_i^t = \sum_{j=1}^N W_{ij} \boldsymbol{s}_j^{t-1} + \tfrac{1}{M}\Big[\nabla_{\boldsymbol{\theta}} J_{i,p_t}(\boldsymbol{\theta}_i^t, \boldsymbol{w}_i^t) - \nabla_{\boldsymbol{\theta}} J_{i,p_t}(\boldsymbol{\theta}_i^{\tau_{p_t}^{t-1}}, \boldsymbol{w}_i^{\tau_{p_t}^{t-1}})\Big], \tag{13}$$

$$\boldsymbol{d}_i^t = \boldsymbol{d}_i^{t-1} + \tfrac{1}{M}\Big[\nabla_{\boldsymbol{w}_i} J_{i,p_t}(\boldsymbol{\theta}_i^t, \boldsymbol{w}_i^t) - \nabla_{\boldsymbol{w}_i} J_{i,p_t}(\boldsymbol{\theta}_i^{\tau_{p_t}^{t-1}}, \boldsymbol{w}_i^{\tau_{p_t}^{t-1}})\Big], \tag{14}$$

   where $\nabla_{\boldsymbol{\theta}} J_{i,p}(\boldsymbol{\theta}_i^0, \boldsymbol{w}_i^0) = \boldsymbol{0}$ and $\nabla_{\boldsymbol{w}_i} J_{i,p}(\boldsymbol{\theta}_i^0, \boldsymbol{w}_i^0) = \boldsymbol{0}$ for all $p \in [M]$ for initialization.
   Perform primal-dual updates using $\boldsymbol{s}_i^t, \boldsymbol{d}_i^t$ as surrogates for the gradients *w.r.t.* $\boldsymbol{\theta}$ and $\boldsymbol{w}_i$:
$$\boldsymbol{\theta}_i^{t+1} = \sum_{j=1}^N W_{ij} \boldsymbol{\theta}_j^t - \gamma_1 \boldsymbol{s}_i^t, \;\; \boldsymbol{w}_i^{t+1} = \boldsymbol{w}_i^t + \gamma_2 \boldsymbol{d}_i^t \;. \tag{15}$$

   **end for**
**end for**

---

maintains a local copy of the primal parameter $\{\boldsymbol{\theta}_i^t\}_{t \geq 1}$. We construct sequences $\{\boldsymbol{s}_i^t\}_{t \geq 1}$ and $\{\boldsymbol{d}_i^t\}_{t \geq 1}$ to track the gradients with respect to $\boldsymbol{\theta}$ and $\boldsymbol{w}_i$, respectively. Similar to (11), in the $t$-th iteration, we update the dual variable via gradient update using $\boldsymbol{d}_i^t$. As for the primal variable, to achieve consensus, each $\boldsymbol{\theta}_i^{t+1}$ is obtained by first combining $\{\boldsymbol{\theta}_i^t\}_{i \in [N]}$ using the weight matrix $\boldsymbol{W}$, and then update in the direction of $\boldsymbol{s}_i^t$. The details of our method are presented in Algorithm 1.

Let us explain the intuition behind the PD-DistIAG method through studying the update (13). Recall that the global gradient desired at iteration $t$ is given by $\nabla_{\boldsymbol{\theta}} J(\boldsymbol{\theta}^t, \{\boldsymbol{w}_i^t\}_{i=1}^N)$, which represents a double average – one over space (across agents) and one over time (across samples). Now in the case of (13), the first summand on the right hand side computes a local average among the neighbors of agent $i$, and thereby tracking the global gradient over *space*. This is in fact akin to the *gradient tracking* technique in the context of distributed optimization [41]. The remaining terms on the right hand side of (13) utilize an incremental update rule akin to the SAG method [43], involving a swap-in swap-out operation for the gradients. This achieves tracking of the global gradient over *time*.

To gain insights on why the scheme works, we note that $\boldsymbol{s}_i^t$ and $\boldsymbol{d}_i^t$ represent some surrogate functions for the primal and dual gradients. Moreover, for the counter variable, using (12) we can alternatively represent it as $\tau_p^t = \max\{\ell \geq 0 : \ell \leq t, \; p_\ell = p\}$. In other words, $\tau_p^t$ is the iteration index where the $p$-th sample is last visited by the agents prior to iteration $t$, and if the $p$-th sample has never been visited, we have $\tau_p^t = 0$. For any $t \geq 1$, define $\boldsymbol{g}_{\boldsymbol{\theta}}(t) := (1/N) \sum_{i=1}^N \boldsymbol{s}_i^t$. The following lemma shows that $\boldsymbol{g}_{\boldsymbol{\theta}}(t)$ is a double average of the primal gradient – it averages over the local gradients across the agents, and for each local gradient; it also averages over the past gradients for all the samples evaluated up till iteration $t+1$. This shows that the average over network for $\{\boldsymbol{s}_i^t\}_{i=1}^N$ can always track the double average of the local and past gradients, *i.e.,* the gradient estimate $\boldsymbol{g}_{\boldsymbol{\theta}}(t)$ is 'unbiased' with respect to the network-wide average.

**Lemma 1** *For all $t \geq 1$ and consider Algorithm 1, it holds that*

$$\boldsymbol{g}_{\boldsymbol{\theta}}(t) = \tfrac{1}{NM} \sum_{i=1}^N \sum_{p=1}^M \nabla_{\boldsymbol{\theta}} J_{i,p}(\boldsymbol{\theta}_i^{\tau_p^t}, \boldsymbol{w}_i^{\tau_p^t}) \;. \tag{16}$$

**Proof.** We shall prove the statement using induction. For the base case with $t = 1$, using (13) and the update rule specified in the algorithm, we have

$$\boldsymbol{g}_{\boldsymbol{\theta}}(1) = \frac{1}{N}\sum_{i=1}^N \frac{1}{M} \nabla_{\boldsymbol{\theta}} J_{i,p_1}(\boldsymbol{\theta}_i^1, \boldsymbol{w}_i^1) = \frac{1}{NM}\sum_{i=1}^N \sum_{p=1}^M \nabla_{\boldsymbol{\theta}} J_{i,p_t}(\boldsymbol{\theta}_i^{\tau_p^1}, \boldsymbol{w}_i^{\tau_p^1}) \;, \tag{17}$$

where we use the fact $\nabla_{\boldsymbol{\theta}} J_{i,p}(\boldsymbol{\theta}_i^{\tau_p^1}, \boldsymbol{w}_i^{\tau_p^1}) = \nabla_{\boldsymbol{\theta}} J_{i,p}(\boldsymbol{\theta}_i^0, \boldsymbol{w}_i^0) = \mathbf{0}$ for all $p \neq p_1$ in the above. For the induction step, suppose (16) holds up to iteration $t$. Since $\boldsymbol{W}$ is doubly stochastic, (13) implies

$$
\begin{aligned}
\boldsymbol{g}_{\boldsymbol{\theta}}(t+1) &= \frac{1}{N} \sum_{i=1}^N \left\{ \sum_{j=1}^N W_{ij} \boldsymbol{s}_j^t + \frac{1}{M} \left[ \nabla_{\boldsymbol{\theta}} J_{i,p_{t+1}}(\boldsymbol{\theta}_i^{t+1}, \boldsymbol{w}_i^{t+1}) - \nabla_{\boldsymbol{\theta}} J_{i,p_{t+1}}(\boldsymbol{\theta}_i^{\tau_{p_{t+1}}^t}, \boldsymbol{w}_i^{\tau_{p_{t+1}}^t}) \right] \right\} \\
&= \boldsymbol{g}_{\boldsymbol{\theta}}(t) + \frac{1}{NM} \sum_{i=1}^N \left[ \nabla_{\boldsymbol{\theta}} J_{i,p_{t+1}}(\boldsymbol{\theta}_i^{t+1}, \boldsymbol{w}_i^{t+1}) - \nabla_{\boldsymbol{\theta}} J_{i,p_{t+1}}(\boldsymbol{\theta}_i^{\tau_{p_{t+1}}^t}, \boldsymbol{w}_i^{\tau_{p_{t+1}}^t}) \right] .
\end{aligned}
\tag{18}
$$

Notice that we have $\tau_{p_{t+1}}^{t+1} = t+1$ and $\tau_p^{t+1} = \tau_p^t$ for all $p \neq p_{t+1}$. The induction assumption in (16) can be written as

$$
\boldsymbol{g}_{\boldsymbol{\theta}}(t) = \frac{1}{NM} \sum_{i=1}^N \left[ \sum_{p \neq p_{t+1}} \nabla_{\boldsymbol{\theta}} J_{i,p}(\boldsymbol{\theta}_i^{\tau_p^{t+1}}, \boldsymbol{w}_i^{\tau_p^{t+1}}) \right] + \frac{1}{NM} \sum_{i=1}^N \nabla_{\boldsymbol{\theta}} J_{i,p_{t+1}}(\boldsymbol{\theta}_i^{\tau_{p_{t+1}}^t}, \boldsymbol{w}_i^{\tau_{p_{t+1}}^t}) . \tag{19}
$$

Finally, combining (18) and (19), we obtain the desired result that (16) holds for the $t + 1$th iteration. This, together with (17), establishes Lemma 1. **Q.E.D.**

As for the dual update (14), we observe the variable $\boldsymbol{w}_i$ is local to agent $i$. Therefore its gradient surrogate, $\boldsymbol{d}_i^t$, involves only the tracking step over *time* [cf. (14)], *i.e.,* it only averages the gradient over samples. Combining with Lemma 1 shows that the PD-DistIAG method uses gradient surrogates that are averages over samples despite the disparities across agents. Since the average over samples are done in a similar spirit as the SAG method, the proposed method is expected to converge linearly.

**Storage and Computation Complexities** Let us comment on the computational and storage complexity of PD-DistIAG method. First of all, since the method requires accessing the previously evaluated gradients, each agent has to store $2M$ such vectors in the memory to avoid re-evaluating these gradients. Each agent needs to store a total of $2Md$ real numbers. On the other hand, the per-iteration computation complexity for each agent is only $\mathcal{O}(d)$ as each iteration only requires to evaluate the gradient over one sample, as delineated in (14)–(15).

**Communication Overhead** The PD-DistIAG method described in Algorithm 1 requires an information exchange round [of $\boldsymbol{s}_i^t$ and $\boldsymbol{\theta}_i^t$] among the agents at every iteration. From an implementation stand point, this may incur significant communication overhead when $d \gg 1$, and it is especially ineffective when the progress made in successive updates of the algorithm is not significant. A natural remedy is to perform multiple *local* updates at the agent using different samples *without* exchanging information with the neighbors. In this way, the communication overhead can be reduced. Actually, this modification to the PD-DistIAG method can be generally described using a time varying weight matrix $\boldsymbol{W}(t)$, such that we have $\boldsymbol{W}(t) = \boldsymbol{I}$ for most of the iteration. The convergence of PD-DistIAG method in this scenario is part of the future work.

### 3.1 Convergence Analysis

The PD-DistIAG method is built using the techniques of (a) primal-dual batch gradient descent, (b) gradient tracking for distributed optimization and (c) stochastic average gradient, where each of them has been independently shown to attain linear convergence under certain conditions; see [41, 43, 20, 15]. Naturally, the PD-DistIAG method is also anticipated to converge at a linear rate.

To see this, let us consider the condition for the sample selection rule of PD-DistIAG:

**A1** *A sample is selected at least once for every $M$ iterations, $|t - \tau_p^t| \leq M$ for all $p \in [M]$, $t \geq 1$.*

The assumption requires that every samples are visited infinitely often. For example, this can be enforced by using a cyclical selection rule, *i.e.,* $p_t = (t \bmod M) + 1$; or a random sampling scheme *without replacement* (*i.e.,* random shuffling) from the pool of $M$ samples. Finally, it is possible to relax the assumption such that a sample can be selected once for every $K$ iterations only, with $K \geq M$. The present assumption is made solely for the purpose of ease of presentation. Moreover, to ensure that the solution to (10) is unique, we consider:

**A2** *The sampled correlation matrix $\hat{\boldsymbol{A}}$ is full rank, and the sampled covariance $\hat{\boldsymbol{C}}$ is non-singular.*

The following theorem confirms the linear convergence of PD-DistIAG:

**Theorem 1** *Under A1 and A2, we denote by $(\boldsymbol{\theta}^\star, \{\boldsymbol{w}_i^\star\}_{i=1}^N)$ the primal-dual optimal solution to the optimization problem in (10). Set the step sizes as $\gamma_2 = \beta\gamma_1$ with $\beta := 8(\rho + \lambda_{\max}(\hat{\boldsymbol{A}}^\top \hat{\boldsymbol{C}}^{-1}\hat{\boldsymbol{A}}))/\lambda_{\min}(\hat{\boldsymbol{C}})$. Define $\overline{\boldsymbol{\theta}}(t) := \frac{1}{N}\sum_{i=1}^N \boldsymbol{\theta}_i^t$ as the average of parameters. If the primal step size $\gamma_1$ is sufficiently small, then there exists a constant $0 < \sigma < 1$ that*

$$\left\|\overline{\boldsymbol{\theta}}(t) - \boldsymbol{\theta}^\star\right\|^2 + (1/\beta N)\sum_{i=1}^N \left\|\boldsymbol{w}_i^t - \boldsymbol{w}_i^\star\right\|^2 = \mathcal{O}(\sigma^t), \quad (1/N)\sum_{i=1}^N \left\|\boldsymbol{\theta}_i^t - \overline{\boldsymbol{\theta}}(t)\right\| = \mathcal{O}(\sigma^t) .$$

*If $N, M \gg 1$ and the graph is geometric, a sufficient condition for convergence is to set $\gamma = \mathcal{O}(1/\max\{N^2, M^2\})$ and the resultant rate is $\sigma = 1 - \mathcal{O}(1/\max\{MN^2, M^3\})$.*

The result above shows the desirable convergence properties for PD-DistIAG method – the primal dual solution $(\overline{\boldsymbol{\theta}}(t), \{\boldsymbol{w}_i^t\}_{i=1}^N)$ converges to $(\boldsymbol{\theta}^\star, \{\boldsymbol{w}_i^\star\}_{i=1}^N)$ at a linear rate; also, the consensual error of the local parameters $\boldsymbol{\theta}_i^t$ converges to zero linearly. A distinguishing feature of our analysis is that it handles the *worst case* convergence of the proposed method, rather than the *expected* convergence rate popular for stochastic / incremental gradient methods.

**Proof Sketch** Our proof is divided into three steps. The first step studies the progress made by the algorithm in one iteration, taking into account the non-idealities due to imperfect tracking of the gradient over space and time. This leads to the characterization of a *Lyapunov vector*. The second step analyzes the *coupled* system of one iteration progress made by the Lyapunov vector. An interesting feature of it is that it consists of a series of independently *delayed* terms in the Lyapunov vector. The latter is resulted from the incremental update schemes employed in the method. Here, we study a sufficient condition for the coupled and delayed system to converge linearly. The last step is to derive condition on the step size $\gamma_1$ where the sufficient convergence condition is satisfied.

Specifically, we study the progress of the Lyapunov functions:

$$\|\widehat{\boldsymbol{v}}(t)\|^2 := \Theta\left(\left\|\overline{\boldsymbol{\theta}}(t) - \boldsymbol{\theta}^\star\right\|^2 + (1/\beta N)\sum_{i=1}^N \left\|\boldsymbol{w}_i^t - \boldsymbol{w}_i^\star\right\|^2\right), \quad \mathcal{E}_c(t) := \frac{1}{N}\sqrt{\sum_{i=1}^N \|\boldsymbol{\theta}_i^t - \overline{\boldsymbol{\theta}}(t)\|^2},$$

$$\mathcal{E}_g(t) := \frac{1}{N}\sqrt{\sum_{i=1}^N \left\|\boldsymbol{s}_i^t - \frac{1}{NM}\sum_{j=1}^N\sum_{p=1}^M \nabla_{\boldsymbol{\theta}} J_{j,p}(\boldsymbol{\theta}_j^{\tau_j^t}, \boldsymbol{w}_j^{\tau_j^t})\right\|^2} .$$

That is, $\widehat{\boldsymbol{v}}(t)$ is a vector whose squared norm is equivalent to a weighted distance to the optimal primal-dual solution, $\mathcal{E}_c(t)$ and $\mathcal{E}_g(t)$ are respectively the consensus errors of the primal parameter and of the primal *aggregated* gradient. These functions form a non-negative vector which evolves as:

$$\begin{pmatrix} \|\widehat{\boldsymbol{v}}(t+1)\| \\ \mathcal{E}_c(t+1) \\ \mathcal{E}_g(t+1) \end{pmatrix} \leq \boldsymbol{Q}(\gamma_1) \begin{pmatrix} \max_{(t-2M)_+\leq q\leq t} \|\widehat{\boldsymbol{v}}(q)\| \\ \max_{(t-2M)_+\leq q\leq t} \mathcal{E}_c(q) \\ \max_{(t-2M)_+\leq q\leq t} \mathcal{E}_g(q) \end{pmatrix} , \qquad (20)$$

where the matrix $\boldsymbol{Q}(\gamma_1) \in \mathbb{R}^{3\times 3}$ is defined by (exact form given in the supplementary material)

$$\boldsymbol{Q}(\gamma_1) = \begin{pmatrix} 1 - \gamma_1 a_0 + \gamma_1^2 a_1 & \gamma_1 a_2 & 0 \\ 0 & \lambda & \gamma_1 \\ \gamma_1 a_3 & a_4 + \gamma_1 a_5 & \lambda + \gamma_1 a_6 \end{pmatrix} . \qquad (21)$$

In the above, $\lambda := \lambda_{\max}(\boldsymbol{W} - (1/N)\mathbf{1}\mathbf{1}^\top) < 1$, and $a_0, ..., a_6$ are some non-negative constants that depends on the problem parameters $N$, $M$, the spectral properties of $\boldsymbol{A}$, $\boldsymbol{C}$, etc., with $a_0$ being positive. If we focus only on the first row of the inequality system, we obtain

$$\|\widehat{\boldsymbol{v}}(t+1)\| \leq \left(1 - \gamma_1 a_0 + \gamma_1^2 a_1\right) \max_{(t-2M)_+\leq q\leq t} \|\widehat{\boldsymbol{v}}(q)\| + \gamma_1 a_2 \max_{(t-2M)_+\leq q\leq t} \mathcal{E}_c(q) .$$

In fact, when the contribution from $\mathcal{E}_c(q)$ can be ignored, then applying [16, Lemma 3] shows that $\|\widehat{\boldsymbol{v}}(t+1)\|$ converges linearly if $-\gamma_1 a_0 + \gamma_1^2 a_1 < 0$, which is possible as $a_0 > 0$. Therefore, if $\mathcal{E}_c(t)$ also converges linearly, then it is anticipated that $\mathcal{E}_g(t)$ would do so as well. In other words, the linear convergence of $\|\widehat{\boldsymbol{v}}(t)\|$, $\mathcal{E}_c(t)$ and $\mathcal{E}_g(t)$ are all coupled in the inequality system (20).

Formalizing the above observations, Lemma 1 in the supplementary material shows a sufficient condition on $\gamma_1$ for linear convergence. Specifically, if there exists $\gamma_1 > 0$ such that the spectral radius of $\boldsymbol{Q}(\gamma_1)$ in (21) is strictly less than one, then each of the Lyapunov functions, $\|\widehat{\boldsymbol{v}}(t)\|$, $\mathcal{E}_c(t)$, $\mathcal{E}_g(t)$, would enjoy linear convergence. Furthermore, Lemma 2 in the supplementary material gives an existence proof for such an $\gamma_1$ to exist. This concludes the proof.

**Remark** While delayed inequality system has been studied in [16, 20] for optimization algorithms, the coupled system in (20) is a non-trivial generalization of the above. Importantly, the challenge here is due to the asymmetry of the system matrix $\boldsymbol{Q}$ and the maximum over the past sequences on the right hand side are taken *independently*. To the best of our knowledge, our result is the first to characterize the (linear) convergence of such coupled and delayed system of inequalities.

**Extension** Our analysis and algorithm may in fact be applied to solve general problems that involves multi-agent and finite-sum optimization, e.g.,

$$\min_{\boldsymbol{\theta} \in \mathbb{R}^d} \ J(\boldsymbol{\theta}) := \frac{1}{NM} \sum_{i=1}^N \sum_{p=1}^M J_{i,p}(\boldsymbol{\theta}) \ . \tag{22}$$

For instance, these problems may arise in multi-agent empirical risk minimization, where data samples are kept independently by agents. Our analysis, especially with convergence for inequality systems of the form (20), can be applied to study a similar double averaging algorithm with just the primal variable. In particular, we only require the sum function $J(\boldsymbol{\theta})$ to be strongly convex, and the objective functions $J_{i,p}(\cdot)$ to be smooth in order to achieve linear convergence. We believe that such extension is of independent interest to the community. At the time of submission, a recent work [40] applied a related double averaging distributed algorithm to a *stochastic version* of (22). However, their convergence rate is sub-linear as they considered a stochastic optimization setting.

## 4 Numerical Experiments

To verify the performance of our proposed method, we conduct an experiment on the `mountaincar` dataset [46] under a setting similar to [15] – to collect the dataset, we ran Sarsa with $d = 300$ features to obtain the policy, then we generate the trajectories of actions and states according to the policy with $M$ samples. For each sample $p$, we generate the local reward, $R_i(s_{p,i}, a_{p,i})$ by assigning a random portion for the reward to each agent such that the average of the local rewards equals $\mathcal{R}_c(\boldsymbol{s}_p, \boldsymbol{a}_p)$.

We compare our method to several centralized methods – PDBG is the primal-dual gradient descent method in (11), GTD2 [47], and SAGA [15]. Notably, SAGA has linear convergence while only requiring an incremental update step of low complexity. For PD-DistIAG, we simulate a communication network with $N = 10$ agents, connected on an Erdos-Renyi graph generated with connectivity of 0.2; for the step sizes, we set $\gamma_1 = 0.005/\lambda_{\max}(\hat{\boldsymbol{A}})$, $\gamma_2 = 5 \times 10^{-3}$.

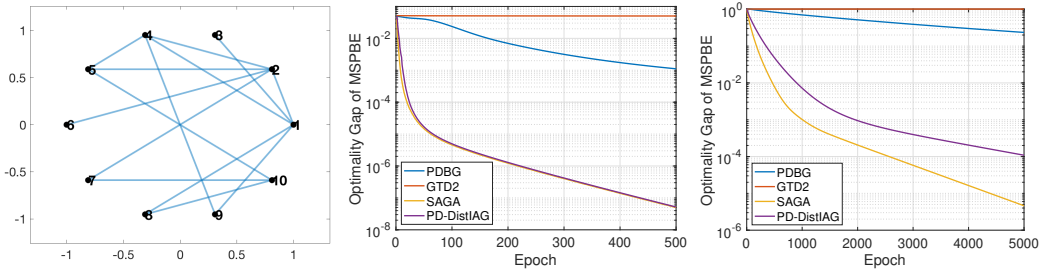

Figure 1: Experiment with `mountaincar` dataset. For this problem, we have $d = 300$, $M = 5000$ samples, and there are $N = 10$ agents. (Left) Graph Topology. (Middle) $\rho = 0.01$. (Right) $\rho = 0$.

Figure 1 compares the optimality gap in terms of MSPBE of different algorithms against the epoch number, defined as $(t/M)$. For PD-DistIAG, we compare its optimality gap in MSPBE as the average objective, *i.e.,* it is $(1/N) \sum_{i=1}^N \mathsf{MSPBE}(\boldsymbol{\theta}_i^t) - \mathsf{MSPBE}(\boldsymbol{\theta}^\star)$. As seen in the left panel, when the regularization factor is high with $\rho > 0$, the convergence speed of PD-DistIAG is comparable to that of SAGA; meanwhile with $\rho = 0$, the PD-DistIAG converges at a slower speed than SAGA. Nevertheless, in both cases, the PD-DistIAG method converges faster than the other methods except for SAGA. Additional experiments are presented in the supplementary materials to compare the performance at different topology and regularization parameter.

**Conclusion** In this paper, we have studied the policy evaluation problem in *multi-agent* reinforcement learning. Utilizing Fenchel duality, a double averaging scheme is proposed to tackle the primal-dual, multi-agent, and finite-sum optimization arises. The proposed PD-DistIAG method demonstrates linear convergence under reasonable assumptions.

**Acknowledgement** The authors would like to thank for the useful comments from three anonymous reviewers. HTW's work was supported by the grant NSF CCF-BSF 1714672. MH's work has been supported in part by NSF-CMMI 1727757, and AFOSR 15RT0767.

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
