[Supplementary Material]

# Supplementary Material for Multi-Agent Reinforcement Learning via Double Averaging Primal-Dual Optimization

**Hoi-To Wai**
The Chinese University of Hong Kong
Shatin, Hong Kong
htwai@se.cuhk.edu.hk

**Zhuoran Yang**
Princeton University
Princeton, NJ, USA
zy6@princeton.edu

**Zhaoran Wang**
Northwestern University
Evanston, IL, USA
zhaoranwang@gmail.com

**Mingyi Hong**
University of Minnesota
Minneapolis, MN, USA
mhong@umn.edu

## A  Proof of Theorem 1

We repeat the statement of the theorem as follows:

**Theorem.** *Under A1 and A2, we denote by $(\boldsymbol{\theta}^\star, \{\boldsymbol{w}_i^\star\}_{i=1}^N)$ the primal-dual optimal solution to the optimization problem in* (10). *Set the step sizes as $\gamma_2 = \beta\gamma_1$ with $\beta := 8(\rho + \lambda_{\max}(\hat{\boldsymbol{A}}^\top \hat{\boldsymbol{C}}^{-1} \hat{\boldsymbol{A}}))/\lambda_{\min}(\hat{\boldsymbol{C}})$. Define $\overline{\boldsymbol{\theta}}(t) := \frac{1}{N}\sum_{i=1}^N \boldsymbol{\theta}_i^t$ as the average of parameters. If the primal step size $\gamma_1$ is sufficiently small, then there exists a constant $0 < \sigma < 1$ that*

$$\left\|\overline{\boldsymbol{\theta}}(t) - \boldsymbol{\theta}^\star\right\|^2 + (1/\beta N)\sum_{i=1}^N \left\|\boldsymbol{w}_i^t - \boldsymbol{w}_i^\star\right\|^2 = \mathcal{O}(\sigma^t), \ \ (1/N)\sum_{i=1}^N \left\|\boldsymbol{\theta}_i^t - \overline{\boldsymbol{\theta}}(t)\right\| = \mathcal{O}(\sigma^t) .$$

*If $N, M \gg 1$ and the graph is geometric with $\lambda = 1 - c/N$ for $c > 0$, a sufficient condition for convergence is to set $\gamma = \mathcal{O}(1/\max\{N^2, M^2\})$ and the resultant rate is $\sigma = 1 - \mathcal{O}(1/\max\{MN^2, M^3\})$.*

**Notation**  We first define a set of notations pertaining to the proof. For any $\beta > 0$, observe that the primal-dual optimal solution, $(\boldsymbol{\theta}^\star, \{\boldsymbol{w}_i^\star\}_{i=1}^N)$, to the optimization problem (10) can be written as

$$\underbrace{\begin{pmatrix} \rho\boldsymbol{I} & \sqrt{\frac{\beta}{N}}\hat{\boldsymbol{A}}^\top & \cdots & \sqrt{\frac{\beta}{N}}\hat{\boldsymbol{A}}^\top \\ -\sqrt{\frac{\beta}{N}}\hat{\boldsymbol{A}} & \beta\hat{\boldsymbol{C}} & \cdots & \cdots \\ \vdots & \boldsymbol{0} & \ddots & \boldsymbol{0} \\ -\sqrt{\frac{\beta}{N}}\hat{\boldsymbol{A}} & \cdots & \cdots & \beta\hat{\boldsymbol{C}} \end{pmatrix}}_{:=\boldsymbol{G}} \begin{pmatrix} \boldsymbol{\theta}^\star \\ \frac{1}{\sqrt{\beta N}}\boldsymbol{w}_1^\star \\ \vdots \\ \frac{1}{\sqrt{\beta N}}\boldsymbol{w}_N^\star \end{pmatrix} = \begin{pmatrix} \boldsymbol{0} \\ -\sqrt{\frac{\beta}{N}}\boldsymbol{b}_1 \\ \vdots \\ -\sqrt{\frac{\beta}{N}}\boldsymbol{b}_N \end{pmatrix}, \qquad \text{(A.1)}$$

where we denote the matrix on the left hand side as $\boldsymbol{G}$. This equation can be obtained by checking the first-order optimality condition. In addition, for any $p \in \{1, \ldots, M\}$, we define the $\boldsymbol{G}_p$ as

$$\boldsymbol{G}_p := \begin{pmatrix} \rho\boldsymbol{I} & \sqrt{\frac{\beta}{N}}\boldsymbol{A}_p^\top & \cdots & \sqrt{\frac{\beta}{N}}\boldsymbol{A}^\top \\ -\sqrt{\frac{\beta}{N}}\boldsymbol{A}_p & \beta\boldsymbol{C}_p & \cdots & \cdots \\ \vdots & \boldsymbol{0} & \ddots & \boldsymbol{0} \\ -\sqrt{\frac{\beta}{N}}\boldsymbol{A}_p & \cdots & \cdots & \beta\boldsymbol{C}_p \end{pmatrix} . \qquad \text{(A.2)}$$

By definition, $\boldsymbol{G}$ is the sample average of $\{\boldsymbol{G}_p\}_{p=1}^M$. Define $\bar{\boldsymbol{\theta}}(t) := (1/N)\sum_{i=1}^N \boldsymbol{\theta}_i^t$ as the average of the local parameters at iteration $t$. Furthermore, we define

$$\boldsymbol{h_\theta}(t) := \rho\bar{\boldsymbol{\theta}}(t) + \frac{1}{N}\sum_{i=1}^N \hat{\boldsymbol{A}}^\top \boldsymbol{w}_i^t, \qquad \boldsymbol{g_\theta}(t) := \frac{1}{NM}\sum_{i=1}^N\sum_{p=1}^M \left(\rho\boldsymbol{\theta}_i^{\tau_p^t} + \boldsymbol{A}_p^\top \boldsymbol{w}_i^{\tau_p^t}\right), \tag{A.3}$$

$$\boldsymbol{h_{w_i}}(t) := \hat{\boldsymbol{A}}\bar{\boldsymbol{\theta}}(t) - \hat{\boldsymbol{C}}\boldsymbol{w}_i^t - \hat{\boldsymbol{b}}_i, \qquad \boldsymbol{g_{w_i}}(t) := \frac{1}{M}\sum_{p=1}^M \left(\boldsymbol{A}_p\boldsymbol{\theta}_i^{\tau_p^t} - \boldsymbol{C}_p\boldsymbol{w}_i^{\tau_p^t} - \boldsymbol{b}_{p,i}\right), \tag{A.4}$$

where $\boldsymbol{h_\theta}(t)$ and $\boldsymbol{h_w}(t) := [\boldsymbol{h_{w_1}}(t), \cdots, \boldsymbol{h_{w_N}}(t)]$ represent the gradients evaluated by a *centralized* and *batch* algorithm. Note that $\boldsymbol{g_\theta}(t)$ defined in (A.3) coincides with that in (16). Using Lemma 1, it can be checked that $\bar{\boldsymbol{\theta}}(t+1) = \bar{\boldsymbol{\theta}}(t) - \gamma_1 \boldsymbol{g_\theta}(t)$ and $\boldsymbol{w}_i^{t+1} = \boldsymbol{w}_i^t - \gamma_2 \boldsymbol{g_{w_i}}(t)$ for all $t \geq 1$. That is, $\bar{\boldsymbol{\theta}}(t+1)$ and $\boldsymbol{w}_i^{t+1}$ can be viewed as primal-dual updates using $\boldsymbol{g_\theta}(t)$ and $\boldsymbol{g_{w_i}}(t)$, which are decentralized counterparts of gradients $\boldsymbol{h_\theta}(t)$ and $\boldsymbol{h_{w_i}}(t)$ defined in (A.3) (A.4).

To simplify the notation, hereafter, we define vectors $\underline{\boldsymbol{h}}(t)$, $\underline{\boldsymbol{g}}(t)$, and $\underline{\boldsymbol{v}}(t)$ by

$$\underline{\boldsymbol{h}}(t) := \begin{pmatrix} \boldsymbol{h_\theta}(t) \\ -\sqrt{\frac{\beta}{N}}\boldsymbol{h_{w_1}}(t) \\ \vdots \\ -\sqrt{\frac{\beta}{N}}\boldsymbol{h_{w_N}}(t) \end{pmatrix}, \underline{\boldsymbol{g}}(t) := \begin{pmatrix} \boldsymbol{g_\theta}(t) \\ -\sqrt{\frac{\beta}{N}}\boldsymbol{g_{w_1}}(t) \\ \vdots \\ -\sqrt{\frac{\beta}{N}}\boldsymbol{g_{w_N}}(t) \end{pmatrix}, \underline{\boldsymbol{v}}(t) := \begin{pmatrix} \bar{\boldsymbol{\theta}}(t) - \boldsymbol{\theta}^\star \\ \frac{1}{\sqrt{\beta N}}(\boldsymbol{w}_1^t - \boldsymbol{w}_1^\star) \\ \vdots \\ \frac{1}{\sqrt{\beta N}}(\boldsymbol{w}_N^t - \boldsymbol{w}_N^\star) \end{pmatrix}.$$
$$\tag{A.5}$$

Using (A.1), it can be verified that (see the detailed derivation in Section A.2)

$$\underline{\boldsymbol{h}}(t) = \boldsymbol{G}\underline{\boldsymbol{v}}(t). \tag{A.6}$$

By adopting the analysis in [15] and under Assumption 2, it can be shown that with

$$\beta := \frac{8(\rho + \lambda_{\mathsf{max}}(\hat{\boldsymbol{A}}^\top \hat{\boldsymbol{C}}^{-1}\hat{\boldsymbol{A}}))}{\lambda_{\mathsf{min}}(\hat{\boldsymbol{C}})},$$

then $\boldsymbol{G}$ is full rank with its eigenvalues satisfying

$$\lambda_{\mathsf{max}}(\boldsymbol{G}) \leq \left|\frac{\lambda_{\mathsf{max}}(\hat{\boldsymbol{C}})}{\lambda_{\mathsf{min}}(\hat{\boldsymbol{C}})}\right|\lambda_{\mathsf{max}}(\rho\boldsymbol{I} + \hat{\boldsymbol{A}}^\top\hat{\boldsymbol{C}}^{-1}\hat{\boldsymbol{A}}), \qquad \lambda_{\mathsf{min}}(\boldsymbol{G}) \geq \frac{8}{9}\lambda_{\mathsf{min}}(\hat{\boldsymbol{A}}^\top\hat{\boldsymbol{C}}^{-1}\hat{\boldsymbol{A}}) > 0. \tag{A.7}$$

Moreover, let $\boldsymbol{G} := \boldsymbol{U}\boldsymbol{\Lambda}\boldsymbol{U}^{-1}$. be the eigen-decomposition of $\boldsymbol{G}$, where $\boldsymbol{\Lambda}$ is a diagonal matrix consists of the eigenvalues of $\boldsymbol{G}$, and the columns of $\boldsymbol{U}$ are the eigenvectors. Then, $\boldsymbol{U}$ is full rank with

$$\|\boldsymbol{U}\| \leq 8(\rho + \lambda_{\mathsf{max}}(\hat{\boldsymbol{A}}^\top\hat{\boldsymbol{C}}^{-1}\hat{\boldsymbol{A}}))\left|\frac{\lambda_{\mathsf{max}}(\hat{\boldsymbol{C}})}{\lambda_{\mathsf{min}}(\hat{\boldsymbol{C}})}\right|, \qquad \|\boldsymbol{U}^{-1}\| \leq \frac{1}{\rho + \lambda_{\mathsf{max}}(\hat{\boldsymbol{A}}^\top\hat{\boldsymbol{C}}^{-1}\hat{\boldsymbol{A}})}. \tag{A.8}$$

Furthermore, we also define the following upper bounds on the spectral norms

$$G := \|\boldsymbol{G}\|, \quad \overline{G} := \max_{p=1,\ldots,M}\|\boldsymbol{G}_p\|, \quad \overline{A} := \max_{p=1,\ldots,M}\|\boldsymbol{A}_p\|, \quad \overline{C} := \max_{p=1,\ldots,M}\|\boldsymbol{C}_p\|. \tag{A.9}$$

Lastly, we define the following two Lyapunov functions

$$\mathcal{E}_c(t) := \frac{1}{N}\sqrt{\sum_{i=1}^N \|\boldsymbol{\theta}_i^t - \overline{\boldsymbol{\theta}}(t)\|^2}, \qquad \mathcal{E}_g(t) := \frac{1}{N}\sqrt{\sum_{i=1}^N \|\boldsymbol{s}_i^t - \boldsymbol{g_\theta}(t)\|^2}. \tag{A.10}$$

Note that we have the following inequalities:

$$\mathcal{E}_c(t) \leq \frac{1}{N}\sum_{i=1}^N \|\boldsymbol{\theta}_i^t - \overline{\boldsymbol{\theta}}(t)\|, \quad \frac{1}{N}\sum_{i=1}^N \|\boldsymbol{\theta}_i^t - \overline{\boldsymbol{\theta}}(t)\| \leq \sqrt{N}\mathcal{E}_c(t), \tag{A.11}$$

which follows from the norm equivalence $\|\boldsymbol{x}\|_2 \leq \|\boldsymbol{x}\|_1 \leq \sqrt{N}\|\boldsymbol{x}\|_2$ for any $\boldsymbol{x} \in \mathbb{R}^N$.

**Convergence Analysis**  We denote that $\gamma_1 = \gamma$ and $\gamma_2 = \beta\gamma$. To study the linear convergence of the PD-DistIAG method, our first step is to establish a bound on the difference from the primal-dual optimal solution, $\underline{v}(t)$. Observe with the choice of our step size ratio,

$$\underline{v}(t+1) = (\boldsymbol{I} - \gamma\boldsymbol{G})\underline{v}(t) + \gamma\big(\underline{h}(t) - \underline{g}(t)\big) . \tag{A.12}$$

Consider the difference vector $\underline{h}(t) - \underline{g}(t)$. Its first block can be evaluated as

$$
\begin{aligned}
\big[\underline{h}(t) - \underline{g}(t)\big]_1 &= \frac{1}{NM} \sum_{i=1}^{N} \sum_{p=1}^{M} \Big( \rho(\bar{\boldsymbol{\theta}}(t) - \boldsymbol{\theta}_i^{\tau_p^t}) + \boldsymbol{A}_p^\top (\boldsymbol{w}_i^t - \boldsymbol{w}_i^{\tau_p^t}) \Big) \\
&= \frac{1}{NM} \sum_{i=1}^{N} \sum_{p=1}^{M} \Big( \rho(\bar{\boldsymbol{\theta}}(t) - \bar{\boldsymbol{\theta}}(\tau_p^t)) + \boldsymbol{A}_p^\top (\boldsymbol{w}_i^t - \boldsymbol{w}_i^{\tau_p^t}) \Big) + \frac{\rho}{NM} \sum_{i=1}^{N} \sum_{p=1}^{M} (\bar{\boldsymbol{\theta}}(\tau_p^t) - \boldsymbol{\theta}_i^{\tau_p^t}) .
\end{aligned}
\tag{A.13}
$$

Meanwhile, for any $i \in \{1, \ldots, N\}$, the $(i+1)$-th block is

$$
\begin{aligned}
\big[\underline{h}(t) - \underline{g}(t)\big]_{i+1} &= \sqrt{\frac{\beta}{N}} \frac{1}{M} \sum_{p=1}^{M} \Big( \boldsymbol{A}_p(\boldsymbol{\theta}_i^{\tau_p^t} - \bar{\boldsymbol{\theta}}(t)) + \boldsymbol{C}_p(\boldsymbol{w}_i^t - \boldsymbol{w}_i^{\tau_p^t}) \Big) \\
&= \sqrt{\frac{\beta}{N}} \frac{1}{M} \sum_{p=1}^{M} \Big( \boldsymbol{A}_p(\bar{\boldsymbol{\theta}}(\tau_p^t) - \bar{\boldsymbol{\theta}}(t)) + \boldsymbol{C}_p(\boldsymbol{w}_i^t - \boldsymbol{w}_i^{\tau_p^t}) \Big) + \sqrt{\frac{\beta}{N}} \frac{1}{M} \sum_{p=1}^{M} \boldsymbol{A}_p(\boldsymbol{\theta}_i^{\tau_p^t} - \bar{\boldsymbol{\theta}}(\tau_p^t)) .
\end{aligned}
\tag{A.14}
$$

For ease of presentation, we stack up and denote the residual terms (related to consensus error) in (A.13) and (A.14) as the vector $\underline{\boldsymbol{\mathcal{E}}}_c(t)$. That is, the first block of $\underline{\boldsymbol{\mathcal{E}}}_c(t)$ is $\rho/(NM) \cdot \sum_{i=1}^{N} \sum_{p=1}^{M} (\bar{\boldsymbol{\theta}}(\tau_p^t) - \boldsymbol{\theta}_i^{\tau_p^t})$, and the remaining blocks are given by $\sqrt{\beta/N} \cdot 1/M \cdot \sum_{p=1}^{M} \boldsymbol{A}_p(\boldsymbol{\theta}_i^{\tau_p^t} - \bar{\boldsymbol{\theta}}(\tau_p^t))$, $\forall i \in \{1, \ldots, N\}$. Then by the definition of $\boldsymbol{G}_p$ in (A.2), we obtain the following simplification:

$$\underline{h}(t) - \underline{g}(t) - \underline{\boldsymbol{\mathcal{E}}}_c(t) = \frac{1}{M} \sum_{p=1}^{M} \boldsymbol{G}_p \Big( \sum_{j=\tau_p^t}^{t-1} \Delta\underline{v}(j) \Big), \tag{A.15}$$

where we have defined

$$\Delta\underline{v}(j) := \begin{pmatrix} \bar{\boldsymbol{\theta}}(j+1) - \bar{\boldsymbol{\theta}}(j) \\ \frac{1}{\sqrt{\beta N}}(\boldsymbol{w}_1^{j+1} - \boldsymbol{w}_1^j) \\ \vdots \\ \frac{1}{\sqrt{\beta N}}(\boldsymbol{w}_N^{j+1} - \boldsymbol{w}_N^j) \end{pmatrix} . \tag{A.16}$$

Clearly, we can express $\Delta\underline{v}(j)$ as $\Delta\underline{v}(j) = \underline{v}(j+1) - \underline{v}(j)$ with $\underline{v}(t)$ defined in (A.5). Combining (A.6) and (A.12), we can also write $\Delta\underline{v}(j)$ in (A.16) as

$$\Delta\underline{v}(j) = \gamma\big[\underline{h}(j) - \underline{g}(j)\big] - \gamma\underline{h}(j) . \tag{A.17}$$

Denoting $\widehat{\underline{v}}(t) := \boldsymbol{U}^{-1}\underline{v}(t)$, multiplying $\boldsymbol{U}^{-1}$ on both sides of (A.12) yields

$$\widehat{\underline{v}}(t+1) = (\boldsymbol{I} - \gamma\boldsymbol{\Lambda})\widehat{\underline{v}}(t) + \gamma\,\boldsymbol{U}^{-1}\big(\underline{h}(t) - \underline{g}(t)\big) . \tag{A.18}$$

Combining (A.15), (A.17), and (A.18), by triangle inequality, we have

$$\|\widehat{\underline{v}}(t+1)\| \leq \tag{A.19}$$

$$\big\|\boldsymbol{I} - \gamma\boldsymbol{\Lambda}\big\|\|\widehat{\underline{v}}(t)\| + \gamma\|\boldsymbol{U}^{-1}\|\bigg\{\|\underline{\boldsymbol{\mathcal{E}}}_c(t)\| + \frac{\gamma\overline{G}}{M} \sum_{p=1}^{M} \sum_{j=\tau_p^t}^{t-1} \big[\|\underline{h}(j)\| + \|\underline{h}(j) - \underline{g}(j)\|\big] \bigg\},$$

where $\overline{G}$ appears in (A.9) and $\underline{\boldsymbol{\mathcal{E}}}_c(t)$ is the residue term of the consensus. Furthermore, simplifying the right-hand side of (A.19) yields

$$
\|\widehat{\underline{\boldsymbol{v}}}(t+1)\| \leq \big\|\boldsymbol{I}-\gamma\boldsymbol{\Lambda}\big\|\|\widehat{\underline{\boldsymbol{v}}}(t)\| + \gamma\|\boldsymbol{U}^{-1}\|\bigg\{\|\underline{\boldsymbol{\mathcal{E}}}_c(t)\| + \gamma\overline{G}\sum_{j=(t-M)_+}^{t-1}\big[\|\underline{\boldsymbol{h}}(j)\| + \|\underline{\boldsymbol{h}}(j)-\underline{\boldsymbol{g}}(j)\|\big]\bigg\}
$$

$$
\leq \big\|\boldsymbol{I}-\gamma\boldsymbol{\Lambda}\big\|\|\widehat{\underline{\boldsymbol{v}}}(t)\| + \gamma\|\boldsymbol{U}^{-1}\|\Bigg(\|\underline{\boldsymbol{\mathcal{E}}}_c(t)\|
$$

$$
+ \gamma\overline{G}\sum_{j=(t-M)_+}^{t-1}\bigg\{\|\underline{\boldsymbol{\mathcal{E}}}_c(j)\| + G\|\boldsymbol{U}\|\|\widehat{\underline{\boldsymbol{v}}}(j)\| + \overline{G}\|\boldsymbol{U}\|\sum_{\ell=(j-M)_+}^{j-1}\big[\|\widehat{\underline{\boldsymbol{v}}}(\ell+1)\| + \|\widehat{\underline{\boldsymbol{v}}}(\ell)\|\big]\bigg\}\Bigg).
$$
$$(A.20)$$

Moreover, using the definition and (A.11), we can upper bound $\|\underline{\boldsymbol{\mathcal{E}}}_c(t)\|$ by

$$
\|\underline{\boldsymbol{\mathcal{E}}}_c(t)\| \leq \frac{1}{M}\sum_{p=1}^{M}\bigg[\big(\rho+\overline{A}\sqrt{\beta N}\big)\frac{1}{N}\sum_{i=1}^{N}\|\boldsymbol{\theta}_i^{\tau_p^t}-\bar{\boldsymbol{\theta}}(\tau_p^t)\|\bigg] \leq \sqrt{N}\big(\rho+\overline{A}\sqrt{\beta N}\big)\max_{(t-M)_+\leq q\leq t}\mathcal{E}_c(q).
$$
$$(A.21)$$

Thus, combining (A.20) and (A.21), we bound $\|\widehat{\underline{\boldsymbol{v}}}(t+1)\|$ by

$$
\|\widehat{\underline{\boldsymbol{v}}}(t+1)\| \leq \big\|\boldsymbol{I}-\gamma\boldsymbol{\Lambda}\big\|\|\widehat{\underline{\boldsymbol{v}}}(t)\| + C_1(\gamma)\max_{(t-2M)_+\leq q\leq t-1}\|\widehat{\underline{\boldsymbol{v}}}(q)\| + C_2(\gamma)\max_{(t-2M)_+\leq q\leq t}\mathcal{E}_c(q),
$$
$$(A.22)$$

where constants $C_1(\gamma)$ and $C_2(\gamma)$ are given by

$$
C_1(\gamma) := \gamma^2\,\|\boldsymbol{U}\|\|\boldsymbol{U}^{-1}\|\overline{G}M\big(G+2\overline{G}M\big),\ C_2(\gamma) := \gamma\|\boldsymbol{U}^{-1}\|\big(1+\gamma\overline{G}M\big)\sqrt{N}\big(\rho+\overline{A}\sqrt{\beta N}\big).
$$

Notice that since $\boldsymbol{U}^{-1}$ is full rank, the squared norm $\|\widehat{\underline{\boldsymbol{v}}}(t)\|^2$ is proportional to $\|\bar{\boldsymbol{\theta}}(t)-\boldsymbol{\theta}^\star\|^2 + (1/\beta N)\sum_{i=1}^{N}\|\boldsymbol{w}_i^\star-\boldsymbol{w}_i^t\|^2$, i.e., the optimality gap at the $t$-th iteration.

We next upper bound $\mathcal{E}_c(t+1)$ as defined in (A.10). Notice that $N\mathcal{E}_c(t+1)$ can be written as Frobenius norm of the matrix $\boldsymbol{\Theta}^{t+1}-\mathbf{1}\bar{\boldsymbol{\theta}}(t+1)^\top$, where $\boldsymbol{\Theta}^{t+1} = ((\boldsymbol{\theta}_1^{t+1})^\top;\cdots;(\boldsymbol{\theta}_N^{t+1})^\top)$. Also, we denote $\boldsymbol{S}^t = ((\boldsymbol{s}_1^t)^\top;\cdots(\boldsymbol{s}_N^t)^\top)$. By the update in (15) and using the triangle inequality, we have

$$
\mathcal{E}_c(t+1) = \frac{1}{N}\|\boldsymbol{\Theta}^{t+1}-\mathbf{1}\bar{\boldsymbol{\theta}}(t+1)^\top\|_F = \frac{1}{N}\|\boldsymbol{W}(\boldsymbol{\Theta}^t-\mathbf{1}\bar{\boldsymbol{\theta}}(t)^\top)-\gamma(\boldsymbol{S}^t-\mathbf{1}\boldsymbol{g}_{\boldsymbol{\theta}}(t)^\top)\|_F
$$
$$
\leq \frac{1}{N}\big(\|\boldsymbol{\Theta}^{t+1}-\mathbf{1}\bar{\boldsymbol{\theta}}(t)^\top\|_F + \gamma\|\boldsymbol{S}^t-\mathbf{1}\boldsymbol{g}_{\boldsymbol{\theta}}(t)^\top\|_F\big).
$$
$$(A.23)$$

Notice that we have $\lambda := \lambda_{\mathsf{max}}(\boldsymbol{W}-(1/N)\mathbf{1}\mathbf{1}^\top) < 1$ as the graph is connected. Using the fact that $N\mathcal{E}_g(t) = \|\boldsymbol{S}^t-\mathbf{1}\boldsymbol{g}_{\boldsymbol{\theta}}(t)^\top\|_F$, the right-hand side of (A.23) can be bounded by

$$
\mathcal{E}_c(t+1) \leq \lambda\,\mathcal{E}_c(t) + \gamma\,\mathcal{E}_g(t),
$$
$$(A.24)$$

where the Lyapunov function $\mathcal{E}_g(t)$ is defined in (A.10).

To conclude the proof, we need to further upper bound $\mathcal{E}_g(t+1)$. To simplify the notation, let us define $\boldsymbol{G}_p^t = (\nabla_{\boldsymbol{\theta}}J_{1,p}(\boldsymbol{\theta}_1^t;\boldsymbol{w}_1^t)^\top;\cdots;\nabla_{\boldsymbol{\theta}}J_{N,p}(\boldsymbol{\theta}_N^t;\boldsymbol{w}_N^t)^\top)$ and observe that

$$
\mathcal{E}_g(t+1) = \frac{1}{N}\Big\|\boldsymbol{S}^{t+1}-\mathbf{1}\boldsymbol{g}_{\boldsymbol{\theta}}(t+1)^\top\Big\|_F = \frac{1}{N}\Big\|\boldsymbol{W}\boldsymbol{S}^t + \tfrac{1}{M}\big(\boldsymbol{G}_{p_{t+1}}^{t+1}-\boldsymbol{G}_{p_{t+1}}^{\tau_{p_{t+1}}^t}\big)-\mathbf{1}\boldsymbol{g}_{\boldsymbol{\theta}}(t+1)^\top\Big\|_F
$$
$$(A.25)$$

where we have used (13). Furthermore, we observe

$$
\mathcal{E}_g(t+1) = \frac{1}{N}\Big\|\boldsymbol{W}(\boldsymbol{S}^t-\mathbf{1}\boldsymbol{g}_{\boldsymbol{\theta}}(t)^\top) + \tfrac{1}{M}\big(\boldsymbol{G}_{p_{t+1}}^{t+1}-\boldsymbol{G}_{p_{t+1}}^{\tau_{p_{t+1}}^t}\big)-\mathbf{1}(\boldsymbol{g}_{\boldsymbol{\theta}}(t+1)-\boldsymbol{g}_{\boldsymbol{\theta}}(t))^\top\Big\|_F
$$
$$
\leq \frac{1}{N}\Big(\|\boldsymbol{W}(\boldsymbol{S}^t-\mathbf{1}\boldsymbol{g}_{\boldsymbol{\theta}}(t)^\top)\|_F + \|\tfrac{1}{M}\big(\boldsymbol{G}_{p_{t+1}}^{t+1}-\boldsymbol{G}_{p_{t+1}}^{\tau_{p_{t+1}}^t}\big)-\mathbf{1}(\boldsymbol{g}_{\boldsymbol{\theta}}(t+1)-\boldsymbol{g}_{\boldsymbol{\theta}}(t))^\top\|_F\Big)
$$
$$
\leq \lambda\mathcal{E}_g(t) + \frac{1}{N}\|\tfrac{1}{M}\big(\boldsymbol{G}_{p_{t+1}}^{t+1}-\boldsymbol{G}_{p_{t+1}}^{\tau_{p_{t+1}}^t}\big)-\mathbf{1}(\boldsymbol{g}_{\boldsymbol{\theta}}(t+1)-\boldsymbol{g}_{\boldsymbol{\theta}}(t))^\top\|_F
$$
$$(A.26)$$

We observe $\boldsymbol{G}_p^t = ((\boldsymbol{w}_1^t)^\top \boldsymbol{A}_p; \cdots ; (\boldsymbol{w}_N^t)^\top \boldsymbol{A}_p) + \rho\,\boldsymbol{\Theta}^t$ and $\boldsymbol{g}_{\boldsymbol{\theta}}(t+1) - \boldsymbol{g}_{\boldsymbol{\theta}}(t) = M^{-1}\big(\rho\,\bar{\boldsymbol{\theta}}(t+1) - \rho\,\bar{\boldsymbol{\theta}}(\tau_{p_{t+1}}^t) + N^{-1}\boldsymbol{A}_{p_{t+1}}^\top \sum_{i=1}^{N}\big(\boldsymbol{w}_i^{t+1} - \boldsymbol{w}_i^{\tau_{p_{t+1}}^t}\big)\big)$. Adopting the notations $\boldsymbol{\Omega}^t = ((\boldsymbol{w}_1^t)^\top; \cdots ; (\boldsymbol{w}_N^t)^\top)$ and $\overline{\boldsymbol{w}}^t = N^{-1}\sum_{i=1}^{N}\boldsymbol{w}_i^t$, we observe that

$$
\begin{aligned}
& M^{-1}(\boldsymbol{G}_{p_{t+1}}^{t+1} - \boldsymbol{G}_{p_{t+1}}^{\tau_{p_{t+1}}^t}) - \mathbf{1}(\boldsymbol{g}_{\boldsymbol{\theta}}(t+1) - \boldsymbol{g}_{\boldsymbol{\theta}}(t))^\top \\
&= \frac{\rho}{M}\big(\boldsymbol{\Theta}^{t+1} - \mathbf{1}\overline{\boldsymbol{\theta}}(t+1)^\top - \boldsymbol{\Theta}^{\tau_{p_{t+1}}^t} - \mathbf{1}\overline{\boldsymbol{\theta}}(\tau_{p_{t+1}}^t)^\top\big) \\
&\quad + \frac{1}{M}\big(\boldsymbol{\Omega}^{t+1} - \mathbf{1}(\overline{\boldsymbol{w}}^{t+1})^\top - \boldsymbol{\Omega}^{\tau_{p_{t+1}}^t} + \mathbf{1}(\overline{\boldsymbol{w}}^{\tau_{p_{t+1}}^t})^\top\big)\boldsymbol{A}_{p_{t+1}}.
\end{aligned}
\tag{A.27}
$$

Using the triangular inequality, the norm of the above can be bounded as

$$
\begin{aligned}
& \frac{\rho}{M}\Big(\|\boldsymbol{\Theta}^{t+1} - \mathbf{1}\overline{\boldsymbol{\theta}}(t+1)^\top\|_F + \|\boldsymbol{\Theta}^{\tau_{p_{t+1}}^t} - \mathbf{1}\overline{\boldsymbol{\theta}}(\tau_{p_{t+1}}^t)^\top\|_F\Big) \\
& + \frac{\|\boldsymbol{A}_{p_{t+1}}\|}{M}\Big(\|\boldsymbol{\Omega}^{t+1} - \mathbf{1}(\overline{\boldsymbol{w}}^{t+1})^\top - \boldsymbol{\Omega}^{\tau_{p_{t+1}}^t} + \mathbf{1}(\overline{\boldsymbol{w}}^{\tau_{p_{t+1}}^t})^\top\|_F\Big)
\end{aligned}
\tag{A.28}
$$

From the norm equivalence $\|\boldsymbol{x}\|_2 \leq \|\boldsymbol{x}\|_1$, we recognize that $\|\boldsymbol{\Omega}^{t+1} - \mathbf{1}(\overline{\boldsymbol{w}}^{t+1})^\top - \boldsymbol{\Omega}^{\tau_{p_{t+1}}^t} + \mathbf{1}(\overline{\boldsymbol{w}}^{\tau_{p_{t+1}}^t})^\top\|_F \leq \sum_{i=1}^{N}\|\boldsymbol{w}_i^{t+1} - \overline{\boldsymbol{w}}^{t+1} - \boldsymbol{w}_i^{\tau_{p_{t+1}}^t} + \overline{\boldsymbol{w}}^{\tau_{p_{t+1}}^t}\|$. It holds for all $t' \leq t$ that

$$
\boldsymbol{w}_i^{t+1} - \boldsymbol{w}_i^{t'} = -\frac{\gamma}{\beta M}\sum_{\ell=t'}^{t}\sum_{p=1}^{M}\Big[\boldsymbol{A}_p(\boldsymbol{\theta}_i^{\tau_p^\ell} - \boldsymbol{\theta}^\star) - \boldsymbol{C}_p(\boldsymbol{w}_i^{\tau_p^\ell} - \boldsymbol{w}_i^\star)\Big].
$$

We thus obtain

$$
\begin{aligned}
& \frac{\|\boldsymbol{A}_{p_{t+1}}\|}{M}\Big(\|\boldsymbol{\Omega}^{t+1} - \mathbf{1}(\overline{\boldsymbol{w}}^{t+1})^\top - \boldsymbol{\Omega}^{\tau_{p_{t+1}}^t} + \mathbf{1}(\overline{\boldsymbol{w}}^{\tau_{p_{t+1}}^t})^\top\|_F\Big) \\
&\leq \frac{\|\boldsymbol{A}_{p_{t+1}}\|}{M}\sum_{i=1}^{N}\Big\|\boldsymbol{w}_i^{t+1} - \frac{1}{N}\sum_{j=1}^{N}\boldsymbol{w}_j^{t+1} - \boldsymbol{w}_i^{\tau_{p_{t+1}}^t} + \frac{1}{N}\sum_{j=1}^{N}\boldsymbol{w}_j^{\tau_{p_{t+1}}^t}\Big\| \\
&\leq \frac{2\gamma\overline{A}}{\beta M^2}\sum_{i=1}^{N}\sum_{\ell=(t-M)_+}^{t}\sum_{p=1}^{M}\Big(\|\boldsymbol{A}_p(\boldsymbol{\theta}_i^{\tau_p^\ell} - \boldsymbol{\theta}^\star) - \boldsymbol{C}_p(\boldsymbol{w}_i^{\tau_p^\ell} - \boldsymbol{w}_i^\star)\|\Big) \\
&\leq \frac{2\gamma\overline{A}}{\beta M}\sum_{i=1}^{N}\sum_{\ell=(t-M)_+}^{t}\Big[\max_{(\ell-M)_+ \leq q \leq \ell}\Big(\overline{A}\|\boldsymbol{\theta}_i^q - \boldsymbol{\theta}^\star\| + \overline{C}\|\boldsymbol{w}_i^q - \boldsymbol{w}_i^\star\|\Big)\Big].
\end{aligned}
\tag{A.29}
$$

Thus, combining (A.24), (A.29), and the definition of $\mathcal{E}_c$ in (A.10), we have

$$
\begin{aligned}
& \frac{1}{N}\|\frac{1}{M}\big(\boldsymbol{G}_{p_{t+1}}^{t+1} - \boldsymbol{G}_{p_{t+1}}^{\tau_{p_{t+1}}^t}\big) - \mathbf{1}(\boldsymbol{g}_{\boldsymbol{\theta}}(t+1) - \boldsymbol{g}_{\boldsymbol{\theta}}(t))^\top\|_F \\
&\leq \frac{\rho}{M}\big[\mathcal{E}_c(\tau_{p_{t+1}}^t) + \mathcal{E}_c(t+1)\big] \\
&\quad + \frac{2\gamma\overline{A}(M+1)}{\beta N M}\sum_{i=1}^{N}\max_{(t-2M)_+ \leq q \leq t}\Big(\overline{A}\|\boldsymbol{\theta}_i^q - \boldsymbol{\theta}^\star\| + \overline{C}\|\boldsymbol{w}_i^q - \boldsymbol{w}_i^\star\|\Big) \\
&\leq \frac{\rho}{M}\big[\mathcal{E}_c(\tau_{p_{t+1}}^t) + \lambda\mathcal{E}_c(t) + \gamma\,\mathcal{E}_g(t)\big] \\
&\quad + \frac{2\gamma\overline{A}(M+1)}{\beta M}\max_{(t-2M)_+ \leq q \leq t}\Big(\overline{A}\,\sqrt{N}\mathcal{E}_c(q) + \overline{A}\,\|\overline{\boldsymbol{\theta}}(q) - \boldsymbol{\theta}^\star\| + \frac{\overline{C}}{N}\sum_{i=1}^{N}\|\boldsymbol{w}_i^q - \boldsymbol{w}_i^\star\|\Big).
\end{aligned}
\tag{A.30}
$$

Combining (A.26) and (A.30), we obtain that

$$
\begin{aligned}
\mathcal{E}_g(t+1) &\leq \Big(\lambda + \frac{\gamma\rho}{M}\Big)\mathcal{E}_g(t) + \Big[\frac{2\gamma\overline{A}^2(M+1)\sqrt{N}}{\beta M} + \frac{2(1+\lambda)}{M}\Big]\max_{(t-2M)_+ \leq q \leq t}\mathcal{E}_c(q) \\
&\quad + \frac{2\gamma\overline{A}(M+1)}{\beta M}\max_{(t-2M)_+ \leq q \leq t}\Big(\overline{A}\,\|\overline{\boldsymbol{\theta}}(q) - \boldsymbol{\theta}^\star\| + \frac{\overline{C}}{N}\sum_{i=1}^{N}\|\boldsymbol{w}_i^q - \boldsymbol{w}_i^\star\|\Big).
\end{aligned}
\tag{A.31}
$$

To bound the last term on the right-hand side of (A.31), For all $q$, we observe that:

$$\left(\overline{A}\,\|\bar{\boldsymbol{\theta}}(q) - \boldsymbol{\theta}^\star\| + \frac{\overline{C}}{N}\sum_{i=1}^{N}\|\boldsymbol{w}_i^q - \boldsymbol{w}_i^\star\|\right)^2$$

$$\leq (N+1)(\overline{A})^2\left[\|\bar{\boldsymbol{\theta}}(q) - \boldsymbol{\theta}^\star\|^2 + \beta\Big(\frac{\overline{C}}{\overline{A}}\Big)^2\frac{1}{\beta N}\sum_{i=1}^{N}\|\boldsymbol{w}_i^q - \boldsymbol{w}_i^\star\|^2\right]$$

$$\leq (N+1)\,\|\boldsymbol{U}\|\max\left\{(\overline{A})^2, \beta(\overline{C})^2\right\}\,\|\underline{\boldsymbol{v}}(q)\|^2\,,$$

which further implies that

$$\mathcal{E}_g(t+1) \leq \Big(\lambda + \frac{\gamma\rho}{M}\Big)\mathcal{E}_g(t) + \Big(\frac{2\gamma\overline{A}^2(M+1)\sqrt{N}}{\beta M} + \frac{2(1+\lambda)}{M}\Big)\max_{(t-2M)_+\leq q\leq t}\mathcal{E}_c(q)$$

$$+ \frac{2\gamma\overline{A}\sqrt{N+1}(M+1)}{\beta M}\,\|\boldsymbol{U}\|\max\{\overline{A}, \sqrt{\beta}\overline{C}\}\max_{(t-2M)_+\leq q\leq t}\|\widehat{\underline{\boldsymbol{v}}}(q)\|\,.$$
(A.32)

Finally, combining (A.22), (A.24), (A.32) shows:

$$\begin{pmatrix} \|\widehat{\underline{\boldsymbol{v}}}(t+1)\| \\ \mathcal{E}_c(t+1) \\ \mathcal{E}_g(t+1) \end{pmatrix} \leq \boldsymbol{Q}\begin{pmatrix} \max_{(t-2M)_+\leq q\leq t}\|\widehat{\underline{\boldsymbol{v}}}(q)\| \\ \max_{(t-2M)_+\leq q\leq t}\mathcal{E}_c(q) \\ \max_{(t-2M)_+\leq q\leq t}\mathcal{E}_g(q) \end{pmatrix}\,,$$
(A.33)

where the inequality sign is applied element-wisely, and $\boldsymbol{Q}$ is a non-negative $3 \times 3$ matrix, defined by:

$$\begin{pmatrix} \theta(\gamma) + \gamma^2\|\boldsymbol{U}\|\|\boldsymbol{U}^{-1}\|\overline{G}M(G + 2\overline{G}M) & \gamma\sqrt{N}\|\boldsymbol{U}\|(1 + \gamma\overline{G}M)(\rho + \overline{A}\sqrt{\beta N}) & 0 \\ 0 & \lambda & \gamma \\ \frac{2\gamma\overline{A}\sqrt{N+1}(M+1)}{\beta M}\,\|\boldsymbol{U}\|\max\{\overline{A}, \sqrt{\beta}\,\overline{C}\} & \sqrt{N}\frac{2\gamma\overline{A}^2(M+1)}{\beta M} + \frac{2(1+\lambda)}{M} & \lambda + \frac{\gamma\rho}{M} \end{pmatrix}\,,$$
(A.34)

where $\theta(\gamma) := \|\boldsymbol{I} - \gamma\boldsymbol{\Lambda}\| = \|\boldsymbol{I} - \gamma\boldsymbol{G}\|$. Note that the upper bounds for $\|\boldsymbol{U}\|$ and $\|\boldsymbol{U}^{-1}\|$ are provided in (A.8). Furthermore, also note that the eigenvalues of $\boldsymbol{G}$ are bounded in (A.7). We could set the stepsize $\gamma$ to be sufficiently small such that such that $\theta(\gamma) := \|\boldsymbol{I} - \gamma\boldsymbol{G}\| < 1$.

Finally, we apply Lemmas 2 and 3 presented in Section A.1 to the recursive inequality in (A.32), which shows that each of $\|\underline{\boldsymbol{v}}(t)\|, \mathcal{E}_c(t), \mathcal{E}_g(t)$ converges linearly with $t$. Therefore, we conclude the proof of Theorem 1.

## A.1 Two Useful Lemmas

In this section, we present two auxiliary lemmas that are used in the proof of Theorem 1. Our first lemma establish the linear convergence of vectors satisfying recursive relations similar to (A.32), provided the spectral radius of $\boldsymbol{Q}$ is less than one. In addition, the second lemma verifies this condition for $\boldsymbol{Q}$ defined in in (A.34).

**Lemma 2.** *Consider a sequence of non-negative vectors* $\{\boldsymbol{e}(t)\}_{t\geq 1} \subseteq \mathbb{R}^n$ *whose evolution is characterized by* $\boldsymbol{e}(t + 1) \leq \boldsymbol{Q}\,\boldsymbol{e}([(t - M + 1)_+, t])$ *for all* $t \geq 1$ *and some fixed integer* $M > 0$, *where* $\boldsymbol{Q} \in \mathbb{R}^{n\times n}$ *is a matrix whose entries are nonnegative, and we define*

$$\boldsymbol{e}(\mathcal{S}) := \begin{pmatrix} \max_{q\in\mathcal{S}}\ e_1(q) \\ \vdots \\ \max_{q\in\mathcal{S}}\ e_n(q) \end{pmatrix} \in \mathbb{R}^n \qquad \text{for any subset } \mathcal{S} \subseteq \mathbb{N}\,.$$

*Moreover, if* $\boldsymbol{Q}$ *irreducible in the sense that there exists an integer* $m$ *such that the entries of* $\boldsymbol{Q}^m$ *are all positive, and the spectral radius of* $\boldsymbol{Q}$, *denoted by* $\rho(\boldsymbol{Q})$, *is strictly less than one, then for any* $t \geq 1$, *we have*

$$\boldsymbol{e}(t) \leq \rho(\boldsymbol{Q})^{\lceil\frac{t-1}{M}\rceil}C_1\boldsymbol{u}_1\,,$$
(A.35)

*where* $\boldsymbol{u}_1 \in \mathbb{R}_{++}^n$ *is the top right eigenvector of* $\boldsymbol{Q}$ *and* $C_1$ *is a constant that depends on the initialization.*

**Proof**. We shall prove the lemma using induction. By the Perron-Frobenius theorem, the eigenvector $\boldsymbol{u}_1$ associated with $\rho(\boldsymbol{Q})$ is unique and is an all-positive vector. Therefore, there exists $C_1$ such that

$$\boldsymbol{e}(1) \leq C_1 \boldsymbol{u}_1 \ . \tag{A.36}$$

Let us first consider the base case with $t = 2, ..., M + 1$, i.e., $\lceil (t - 1)/M \rceil = 1$. When $t = 2$, by (A.36) we have,

$$\boldsymbol{e}(2) \leq \boldsymbol{Q}\boldsymbol{e}(1) \leq C_1 \boldsymbol{Q}\boldsymbol{u}_1 = \rho(\boldsymbol{Q})C_1 \boldsymbol{u}_1 \ , \tag{A.37}$$

which is valid as $\boldsymbol{Q}, \boldsymbol{e}(1), \boldsymbol{u}_1$ are all non-negative. Furthermore, we observe that $\boldsymbol{e}(2) \leq C_1 \boldsymbol{u}_1$. Next when $t = 3$, we have

$$\boldsymbol{e}(3) \leq \boldsymbol{Q}\boldsymbol{e}([1,2]) \overset{(a)}{\leq} C_1 \boldsymbol{Q}\boldsymbol{u}_1 = \rho(\boldsymbol{Q})C_1 \boldsymbol{u}_1 \ ,$$

where (a) is due to the non-negativity of vectors/matrix and the fact $\boldsymbol{e}(1), \boldsymbol{e}(2) \leq C_1 \boldsymbol{u}_1$ as shown in (A.37). Telescoping using similar steps, one can show $\boldsymbol{e}(t) \leq \rho(\boldsymbol{Q})C_1 \boldsymbol{u}_1$ for any $t = 2, ..., M + 1$.

For the induction step, let us assume that (A.35) holds true for any $t$ up to $t = pM + 1$. That is, we assume that the result holds for all $t$ such that $\lceil (t - 1)/M \rceil \leq p$. We shall show that it also holds for any $t = pM + 2, ..., (p + 1)M + 1$, i.e., $\lceil (t - 1)/M \rceil = p + 1$. Observe that

$$\boldsymbol{e}(pM + 2) \leq \boldsymbol{Q}\boldsymbol{e}([(p - 1)M + 2, pM + 1]) \leq C_1 \rho(\boldsymbol{Q})^p \boldsymbol{Q}\boldsymbol{u}_1 = \rho(\boldsymbol{Q})^{p+1} C_1 \boldsymbol{u}_1 \ , \tag{A.38}$$

where we have used the induction hypothesis. It is clear that (A.38) is equivalent to (A.35) with $t = pM + 2$. Similar upper bound can be obtained for $\boldsymbol{e}(pM + 3)$ as well. Repeating the same steps, we show that (A.35) is true for any $t = pM + 2, ..., (p + 1)M + 1$. Therefore, we conclude the proof of this lemma.                                                                                                    **Q.E.D.**

The following Lemma shows that $\boldsymbol{Q}$ defined in (A.34) satisfies the conditions required in the previous lemma. Combining these two lemmas yields the final step of the proof of Theorem 1.

**Lemma 3.** *Consider the matrix $\boldsymbol{Q}$ defined in (A.34), it can be shown that (a) $\boldsymbol{Q}$ is an irreducible matrix in $\mathbb{R}^{3 \times 3}$; (b) there exists a sufficiently small $\gamma$ such that $\rho(\boldsymbol{Q}) < 1$; and (c) as $N, M \gg 1$ and the graph is geometric, we can set $\gamma = \mathcal{O}(1/\max\{N^2, M^2\})$ and $\rho(\boldsymbol{Q}) \leq 1 - \mathcal{O}(1/\max\{N^2, M^2\})$.*

**Proof.** Our proof is divided into three parts. The first part shows the straightforward irreducibility of $\boldsymbol{Q}$; the second part gives an upper bound to the spectral radius of $\boldsymbol{Q}$; and the last part derives an asymptotic bound on $\rho(\boldsymbol{Q})$ when $N, M \gg 1$.

**Irreducibility of $\boldsymbol{Q}$**    To see that $\boldsymbol{Q}$ is irreducible, notice that $\boldsymbol{Q}^2$ is a positive matrix, which could be verified by direct computation.

**Spectral Radius of $\boldsymbol{Q}$**    In the sequel, we compute an upper bound to the spectral radius of $\boldsymbol{Q}$, and show that if $\gamma$ is sufficiently small, then its spectral radius will be strictly less than one. First we note that $\theta(\gamma) = 1 - \gamma\alpha$ for some $\alpha > 0$ and the network connectivity satisfies $\lambda < 1$. Also note that $\rho > 0$. For notational simplicity let us define the following

$$a_1 = \|\boldsymbol{U}\|\|\boldsymbol{U}^{-1}\|\overline{G}M(G + 2\overline{G}M), \quad a_2 = \|\boldsymbol{U}\|\sqrt{N}(\rho + \overline{A}\sqrt{\beta N}), \quad a_3 = \overline{G}M\|\boldsymbol{U}\|\sqrt{N}(\rho + \overline{A}\sqrt{\beta N})$$

$$a_4 = \frac{2\overline{A}\sqrt{N + 1}(M + 1)}{\beta M}\|\boldsymbol{U}\|\max\{\overline{A}, \sqrt{\beta\overline{C}}\}, \quad a_5 = \frac{2\overline{A}^2(M + 1)\sqrt{N}}{\beta M}, \quad a_6 = \frac{2(1 + \lambda)}{M}.$$

With the above shorthand definitions, the characteristic polynomial for $\boldsymbol{Q}$, denoted by $g \colon \mathbb{R} \to \mathbb{R}$, is given by

$$g(\sigma) = \det \begin{pmatrix} \sigma - (1 - \gamma\alpha + \gamma^2 a_1) & -\gamma a_2 - \gamma^2 a_3 & 0 \\ 0 & \sigma - \lambda & -\gamma \\ -\gamma a_4 & -\gamma a_5 - a_6 & \sigma - \left(\lambda + \frac{\gamma\rho}{M}\right) \end{pmatrix}.$$

By direct computation, we have

$$g(\sigma) = (\sigma - (1 - \gamma\alpha + \gamma^2 a_1))g_0(\sigma) - \gamma^3(a_2 + \gamma a_3)a_4 \tag{A.39}$$

where

$$g_0(\sigma) := (\sigma - \lambda)^2 - \frac{\gamma\rho}{M}(\sigma - \lambda) - \gamma(\gamma a_5 + a_6) . \tag{A.40}$$

Notice that the two roots of the above polynomial can be upper bounded by:

$$\lambda + \frac{\gamma\rho}{2M} \pm \sqrt{\left(\frac{\gamma\rho\sqrt{N}}{2M}\right)^2 + \gamma(\gamma a_5 + a_6)} \leq \overline{\sigma} := \lambda + \frac{\gamma\rho}{M} + \sqrt{\gamma(\gamma a_5 + a_6)} \tag{A.41}$$

In particular, for all $\sigma \geq \overline{\sigma}$, we have

$$g_0(\sigma) \geq (\sigma - \overline{\sigma})^2 . \tag{A.42}$$

Now, let us define

$$\sigma^\star := \max\left\{ \frac{\gamma\alpha}{4} + 1 - \gamma\alpha + \gamma^2 a_1, \overline{\sigma} + \gamma\sqrt{\frac{4(a_2 + \gamma a_3)a_4}{\alpha}} \right\} \tag{A.43}$$

Observe that for all $\sigma \geq \sigma^\star$, it holds that

$$\begin{aligned}
g(\sigma) &\geq (\sigma - (1 - \gamma\alpha + \gamma^2 a_1))(\sigma - \overline{\sigma})^2 - \gamma^3(a_2 + \gamma a_3)a_4 \\
&\geq \frac{\gamma\alpha}{4}\gamma^2 \frac{4(a_2 + \gamma a_3)a_4}{\alpha} - \gamma^3(a_2 + \gamma a_3)a_4 = 0 .
\end{aligned} \tag{A.44}$$

Lastly, observe that $g(\sigma)$ is strictly increasing for all $\sigma \geq \sigma^\star$. Combining with the Perron Frobenius theorem shows that $\rho(\boldsymbol{Q}) \leq \sigma^\star$. Moreover, as $\lambda < 1$ and $\alpha > 0$, there exists a sufficiently small $\gamma$ such that $\sigma^\star < 1$. We conclude that $\rho(\boldsymbol{Q}) < 1$ in the latter case.

**Asymptotic Rate when** $M, N \gg 1$   We evaluate a sufficient condition on $\gamma$ for the proposed algorithm to converge, *i.e.,* when $\sigma^\star < 1$. Let us consider (A.43) and the first operand in the $\max\{\cdot\}$. The first operand is guaranteed to be less than one if:

$$\gamma \leq \frac{\alpha}{2a_1} \implies \frac{\gamma\alpha}{4} + 1 - \gamma\alpha + \gamma^2 a_1 \leq 1 - \frac{\gamma\alpha}{4} . \tag{A.45}$$

Moreover, from the definition of $a_1$, we note that this requires $\gamma = \mathcal{O}(1/M^2)$ if $M \gg 1$.

Next, we notice that for geometric graphs, we have $\lambda = 1 - c/N$ for some positive $c$. Substituting this into the second operand in (A.43) gives

$$1 - \frac{c}{N} + \frac{\gamma\rho}{M} + \sqrt{\gamma(\gamma a_5 + a_6)} + \gamma\sqrt{\frac{4(a_2 + \gamma a_3)a_4}{\alpha}} < 1 . \tag{A.46}$$

Therefore, (A.45) and (A.46) together give a sufficient condition for $\sigma^\star < 1$.

To obtain an asymptotic rate when $M, N \gg 1$. Observe that $a_2 = \Theta(N)$, $a_3 = \Theta(NM)$, $a_4 = \Theta(\sqrt{N})$, $a_5 = \Theta(\sqrt{N})$, $a_6 = \Theta(1/M)$. Moreover, the condition (A.45) gives $\gamma = \mathcal{O}(1/M^2)$, therefore the left hand side of Eq. (A.46) can be approximated by

$$1 - \frac{c}{N} + \gamma\Theta\left(N^{\frac{3}{4}}\right) + \sqrt{\gamma}\Theta(1/\sqrt{M}) \tag{A.47}$$

Setting the above to $1 - c/(2N)$ requires one to have $\gamma = \mathcal{O}(1/N^2)$.

Finally, the above discussions show that setting $\gamma = \mathcal{O}(1/\max\{N^2, M^2\})$ guarantees that $\sigma^\star < 1$. In particular, we have $\sigma^\star \leq \max\{1 - \gamma\frac{\alpha}{4}, 1 - c/(2N)\} = 1 - \mathcal{O}(1/\max\{N^2, M^2\})$   **Q.E.D.**

## A.2   Derivation of Equation (A.6)

We we establish (A.6) with details. Recall that $\underline{h}(t)$ and $\underline{v}(t)$ are defined in (A.5). We verify this equation for each block of $\underline{h}(t)$. To begin with, for the first block, for $\boldsymbol{h_\theta}(t)$ defined in (A.3), we have

$$\boldsymbol{h_\theta}(t) = \rho\bar{\boldsymbol{\theta}}(t) + \frac{1}{N}\sum_{i=1}^{N}\hat{\boldsymbol{A}}^\top \boldsymbol{w}_i^t = \rho\big(\bar{\boldsymbol{\theta}}(t) - \boldsymbol{\theta}^\star + \boldsymbol{\theta}^\star\big) + \frac{1}{N}\sum_{i=1}^{N}\hat{\boldsymbol{A}}^\top \boldsymbol{w}_i^t .$$

Recall from (A.1) that $\rho\boldsymbol{\theta}^\star = -\frac{1}{N}\sum_{i=1}^{N}\hat{\boldsymbol{A}}^\top\boldsymbol{w}_i^\star$, which implies that

$$\boldsymbol{h}_{\boldsymbol{\theta}}(t) = \rho\big(\bar{\boldsymbol{\theta}}(t) - \boldsymbol{\theta}^\star\big) + \sum_{i=1}^{N}\sqrt{\frac{\beta}{N}}\hat{\boldsymbol{A}}^\top\frac{1}{\sqrt{\beta N}}\big(\boldsymbol{w}_i^t - \boldsymbol{w}_i^\star\big) = [\boldsymbol{G}\underline{\boldsymbol{v}}(t)]_1 \, , \qquad (A.48)$$

where $[\boldsymbol{G}\underline{\boldsymbol{v}}(t)]_1$ denotes the first block of $\boldsymbol{G}\underline{\boldsymbol{v}}(t)$.

It remains to establish the equation for the remaining blocks. For any $i \in \{1, \ldots, N\}$, let us focus on the $i+1$-th block. By the definition of $\boldsymbol{h}_{\boldsymbol{w}_i}(t)$ in (A.4), we have

$$-\sqrt{\frac{\beta}{N}}\boldsymbol{h}_{\boldsymbol{w}_i}(t) = -\sqrt{\frac{\beta}{N}}\big(\hat{\boldsymbol{A}}\bar{\boldsymbol{\theta}}(t) - \hat{\boldsymbol{C}}\boldsymbol{w}_i^t - \hat{\boldsymbol{b}}_i\big) = -\sqrt{\frac{\beta}{N}}\big(\hat{\boldsymbol{A}}(\bar{\boldsymbol{\theta}}(t) - \boldsymbol{\theta}^\star) + \hat{\boldsymbol{A}}\boldsymbol{\theta}^\star - \hat{\boldsymbol{C}}\boldsymbol{w}_i^t - \hat{\boldsymbol{b}}_i\big) \, .$$

Again from (A.1), it holds that $\hat{\boldsymbol{A}}\boldsymbol{\theta}^\star = \boldsymbol{b}_i + \hat{\boldsymbol{C}}\boldsymbol{w}_i^\star$. Therefore,

$$-\sqrt{\frac{\beta}{N}}\big(\hat{\boldsymbol{A}}\bar{\boldsymbol{\theta}}(t) - \hat{\boldsymbol{C}}\boldsymbol{w}_i^t - \hat{\boldsymbol{b}}_i\big) = -\sqrt{\frac{\beta}{N}}\hat{\boldsymbol{A}}(\bar{\boldsymbol{\theta}}(t) - \boldsymbol{\theta}^\star) + \beta\hat{\boldsymbol{C}}\frac{\boldsymbol{w}_i^t - \boldsymbol{w}_i^\star}{\sqrt{\beta N}} = [\boldsymbol{G}\underline{\boldsymbol{v}}(t)]_{i+1} \, , \quad (A.49)$$

where $[\boldsymbol{G}\underline{\boldsymbol{v}}(t)]_{i+1}$ denotes the $i+1$-th block of $\boldsymbol{G}\underline{\boldsymbol{v}}(t)$. Combining (A.48) and (A.49) gives the desired equality.

# B   Additional Experiments

An interesting observation from Theorem 1 is that the convergence rate of PD-DistIAG depends on $M$ and the topology of the graph. The following experiments will demonstrate the effects of these on the algorithm, along with the effects of regularization parameter $\rho$.

Figure B.1: Illustrating the graph topologies in the additional experiments. (Left) ER graph with connectivity probability of $1.01 \log N/N$. (Right) Ring graph.

Figure B.2: Experiment with `mountaincar` dataset. For this problem, we only have $d = 300$, $M = 500$ samples, but yet there are $N = 500$ agents. (Left) We set $\rho = 0.01$. (Right) We set $\rho = 0.1$.

To demonstrate the dependence of PD-DistIAG on the graph topology, we fix the number of agents at $N = 500$ and compare the performances on the ring and the ER graph set with probability of

connection of $p = 1.01 \log N/N$, as illustrated in Fig. B.1. Notice that the ring graph is not a geometric graph and its connectivity parameter, defined as $\lambda := \lambda_{\max}(\boldsymbol{W} - (1/N)\mathbf{1}\mathbf{1}^\top)$ from the previous section can be much closer to 1 than the ER graph. Therefore, we expect the PD-DistIAG algorithm to converge slower on the ring graph. This is corroborated by Fig. B.2. Furthermore, from the figure, we observe that with a larger regularization $\rho$, the disadvantage for using the ring graph has exacerbated. We suspect that this is due to the fact that the convergence speed is limited by the graph connectivity, as seen in (A.34); while in the case of ER graph, the algorithm is able to exploit the improved problem's condition number.

Next, we consider the same set of experiment but increase the number of samples to $M = 5000$.

Figure B.3: Experiment with `mountaincar` dataset. For this problem, we have $d = 300$, $M = 5000$ samples, but yet there are $N = 500$ agents. (Left) We set $\rho = 0.01$. (Right) We set $\rho = 0.1$.

Interestingly, for this example, the performances of the ring graph and the ER graph settings are almost identical in this setting with large sample size $M$. This is possible as we recall from Theorem 1 that the algorithm converges at a rate of $\mathcal{O}(\sigma^t)$ where $\sigma = 1 - \mathcal{O}(1/\max\{MN^2, M^3\})$. As we have $M \gg N$, the impact from the sample size $M$ becomes dominant, and is thus insensitive to the graph's connectivity.