[Reviews · NeurIPS 2018]

Reviewer 1



Summary: This paper studies a policy evaluation problem in distributed cooperative multi-agent learning setting. The authors proposed a double average scheme that converges to the optimal solution at a linear rate. The experiments results show the proposed decentralized algorithm can achieve nearly the same performance as the best centralized baseline (SAGA) and better than that of the other three centralized baselines. Methodology: - As the authors mentioned, theorem 1 doesn’t explicitly specify the step size and convergence rate, which leads to relatively weak results from Theorem 1. I am curious how does different step size affect the convergence. Intuitively we cannot expect the algorithm converge properly if we choose the step size too large/small. - Relation to the centralized algorithms. Is it possible that this algorithm connects to certain centralized algorithms when it is a complete graph? Writing: - This paper is very well-written. - There are typos in Line 135, Line 413. Experiments: The experiments were not sufficiently conducted and discussed. - Could the authors show more than two variants on \rho and compare the performance? What if the \rho becomes larger and how will this affect the performance of the proposed method? - What’s the reason that the proposed algorithm can perform as good as a centralized algorithm when \rho = 0? Could the authors provide some explanations on this? - Will the algorithm perform same as the SAGA even for different \rho when you have a complete graph? I am curious if the connectivity how the performance changes as the connectivity increases.

Reviewer 2



The article extends previous work of primal-dual optimisation for policy evaluation in RL to the distributed policy evaluation setting, maintaining attractive convergence rates for the extended algorithm. Overall, the article gradually builds its contribution and is reasonably easy to follow. A few exception to this are the start of related work, dropping citations in lists, and the lack of an explanation of the repeatedly mentioned 'convex-concave saddle-point problem'. The authors equate averaging over 'agents' with averaging over 'space', which is somewhat of an imprecise metaphorical stretch in my view. The contribution is honestly delineated (collaborative distributed policy evaluation with local rewards), and relevant related work is cited clearly. However, the innovation beyond related work is limited (averaging over agents is added). Empirical results show an improvement over two baselines, but a lag behind SAGA. From my understanding that algorithm has full central information, and is therefore given to indicate a sort of gap between the decentralised solution vs a centralised solver. This is not clear from the text. Finally, the conclusion is called 'discussions' and has several language mistakes, which together with the shallow comments on the empirical performance puts an unsatisfactory end to this otherwise pleasant to read paper. Minor remarks: - 'settings with batch trajectory' -> -ies - Assumption names like 'A1' are easy to miss, and might better be written out in full. Also, next to the first mention of A1 the citation is empty. - J(w, \theta) and J(\theta, w) are inconsistently used around Equation 12. - I could not spot what H1 refers to, when it is used after Equation 13. Maybe A1?

Reviewer 3



This paper proposed a multi-agent reinforcment learning framework by a double averaging scheme where each agent performs averaging over both space (neighbors) and time to incorporate gradient and local reward. The basic idea to achieve this is to reformulate the task in a primal-dual way to a decentralized convex-concave saddle-point problem. Authors also prove that the proposed algorithm achieve fast finite-time convergence. The paper is technically sound but the presentation of the paper need improvement. Some notations are ambitious and over complicated, making the equations difficult to follow. e.g. "s" is used as the state variable in the early sections of the paper but in Eq. 14, "s" is used as the surrogate for the gradient w.r.t. \theta. Are they the same variable? If so, why one can use the state as the surrogates for the gradient w.r.t. \theta. The experiment is weak where a network with only 10 agents are simulated. Can authors include more experiments with a large scale?